# Differentially Private Online-to-Batch for Smooth Losses

**Qinzi Zhang**
Dept. of Electrical and Computer Engineering
Boston University
qinziz@bu.edu

**Hoang Tran**
Dept. of Electrical and Computer Engineering
Boston University
tranhp@bu.edu

**Ashok Cutkosky**
Dept. of Electrical and Computer Engineering
Boston University
ashok@cutkosky.com

## Abstract

We develop a new reduction that converts any online convex optimization algorithm suffering $O(\sqrt{T})$ regret into an $\epsilon$-differentially private stochastic convex optimization algorithm with the optimal convergence rate $\tilde{O}(1/\sqrt{T} + \sqrt{d}/\epsilon T)$ on smooth losses in linear time, forming a direct analogy to the classical non-private "online-to-batch" conversion. By applying our techniques to more advanced adaptive online algorithms, we produce adaptive differentially private counterparts whose convergence rates depend on apriori unknown variances or parameter norms.

## 1 Introduction

Solving stochastic convex optimization (SCO) problems forms a core component of training models in machine learning, and is the topic of this paper. The SCO problem is to optimize an objective $\mathcal{L}$:

$$\min_{x \in W} \mathcal{L}(x) = \mathbb{E}_{z \sim P_z} [\ell(x, z)] \tag{1}$$

Here, $x$ represents model parameters or weights lying in a convex domain $W \subset \mathbb{R}^d$, $P_z$ is some unknown distribution over examples $z$ and $\ell(x, z)$ represents a loss function we will assume to be convex and smooth in $x$. Although we do not know $P_z$, we do know the loss $\ell$, and we have access to an i.i.d. dataset $Z = (z_1, \ldots, z_T)$ that may have been obtained via user surveys or volunteer tests. Using this information, we would like to solve the optimization problem (1). The quality of a putative solution $\hat{x}$ is measured by the *suboptimality* gap $\mathcal{L}(\hat{x}) - \mathcal{L}(x_\star)$ for $x_\star \in \operatorname{argmin}_{x \in W} \mathcal{L}(x)$.

In addition to solving (1), we also wish to *preserve privacy* for the people who contributed to the dataset $Z$. To this end, we require our algorithms to be *differentially private* (Dwork et al., 2006; Dwork and Roth, 2014), which means that replacing any one $z_t$ with a different $z'_t$ has a negligible effect on $\hat{x}$. There is a delicate interplay between privacy, dataset size, and solution quality. As $T$ grows, the influence of any individual $z_t$ on $\hat{x}$ decreases, increasing privacy. However, for any finite $T$, one must necessarily leak *some* information about the dataset in order to achieve a nontrivial $\mathcal{L}(\hat{x}) - \mathcal{L}(x_\star)$. The goal, then, is to minimize $\mathcal{L}(\hat{x}) - \mathcal{L}(x_\star)$ subject to the privacy constraint.

This problem has been well-studied in the literature, and by now the optimal tradeoffs are known and achievable[1]. In particular, Bassily et al. (2019) exhibits a polytime algorithm that achieves

---

[1]We focus on the harder stochastic optimization problem rather than empirical risk minimization, which is also well-studied, e.g. (Chaudhuri et al., 2011; Kifer et al., 2012).

$\mathcal{L}(\hat{x}) - \mathcal{L}(x_\star) \leq \tilde{O}(1/\sqrt{T} + \sqrt{d}/\epsilon T)$ where $\epsilon$ is a measure of privacy loss (large $\epsilon$ means less private), and moreover they show that this bound is tight in the worst case. Then, Feldman et al. (2020) provides an improved algorithm that obtains the same guarantee in $O(T)$ time. Further work on this problem considers assumptions on the geometry (Asi et al., 2021), gradient distributions (Kamath et al., 2021), or smoothness (Bassily et al., 2020; Kulkarni et al., 2021).

All private optimization algorithms we are aware of fall into one of two camps: either they employ some simple pre-processing that "sanitizes" the inputs to a non-private optimization algorithm (e.g. the empirically popular DP-SGD of Abadi et al. (2016)), or they make a careful analysis of the dynamics of their algorithm (e.g. bounding the sensitivity of a single step of stochastic gradient descent). In the former case, the algorithm typically does *not* achieve the optimal convergence rate for stochastic optimization. In the latter case, the algorithm becomes more rigidly tied to the privacy analysis, resulting in delicate "theory-crafted" methods that are less popular in practice.

In contrast, in the non-private setting, there is a simple and general technique to produce stochastic optimization algorithms with optimal convergence guarantees: the *online-to-batch conversion* (Cesa-Bianchi et al., 2004). This method directly converts any *online convex optimization* (OCO)(Shalev-Shwartz, 2011; Hazan, 2019; Orabona, 2019) algorithm into a stochastic optimization algorithm. OCO is a simple and elegant game-theoretic formulation of the optimization problem that has witnessed an explosion of diverse algorithms and techniques, so that this conversion result immediately implies a vast array of practical optimization algorithms. In summary: producing *private* stochastic optimization algorithms with optimal convergence rates is currently delicate and difficult, while producing non-private algorithms is essentially trivial.

Our goal is to rectify this issue. To do so, we produce a new *differentially private* online-to-batch conversion. In direct analogy to the non-private conversion, our method converts any OCO algorithm into a *private* stochastic optimization algorithm. After using our conversion, any OCO algorithm that achieves the optimal regret (the standard measure of algorithm quality in OCO), will automatically achieve the optimal suboptimality gap of $\tilde{O}(1/\sqrt{T}+\sqrt{d}/\epsilon T)$. Our conversion has additional desirable properties: although convexity is required for the guarantee on suboptimality, it is *not* required for the privacy guarantee, meaning that the method can be easily applied to non-convex tasks (e.g. in deep learning). Further, by largely decoupling the privacy analysis from the algorithm design through this reduction, we can leverage the rich literature of OCO to build private algorithms with new *adaptive* guarantees. For two explicit examples, (1) we construct an algorithm guaranteeing $\mathcal{L}(\hat{x}) - \mathcal{L}(x_\star) \leq \tilde{O}(\sigma/\sqrt{T} + \sqrt{d}/\epsilon T)$, where $\sigma$ is the apriori unknown standard deviation in the gradient $\nabla\ell(w,z)$, and (2), we construct an algorithm guaranteeing $\mathcal{L}(\hat{x}) - \mathcal{L}(x_\star) \leq \tilde{O}(\|x_\star - x_1\|/\sqrt{T} + \sqrt{d}/\epsilon T)$, where $x_1$ is any user-supplied point. This last guarantee may have applications in private *fine tuning* (Li et al., 2022; Yu et al., 2021; Hoory et al., 2021; Kurakin et al., 2022; Mehta et al., 2022), as the bound automatically improves when the algorithm is provided with a good initialization point.

## 1.1 Preliminaries

**Problem setup**   We define the loss function as $\ell : W \times Z \to \mathbb{R}$ where $W \subseteq \mathbb{R}^d$ is a convex domain and $Z = (z_1, \ldots, z_T)$ is an element of $\mathcal{Z}^T$ for some dataspace $\mathcal{Z}$. We assume $Z$ is an i.i.d. dataset and $z_t \sim P_z$ for some unknown disribution over $\mathcal{Z}$. We then define $\mathcal{L}(x) = \mathbb{E}_{z\sim P_z}[\ell(x,z)]$.

Let $\|\cdot\|$ be a norm on $\mathbb{R}^d$, with dual norm $\|\cdot\|_*$ defined by $\|g\|_* = \sup_{\|x\|\leq 1}\langle g, x\rangle$. By definition, $\langle g, x\rangle \leq \|g\|_*\|x\|$, (*Fenchel-Young's inequality*). We make the following assumptions:

**Assumption 1.** $\|\cdot\|^2$ is $\lambda$-strongly convex w.r.t. $\|\cdot\|$: for all $x, y \in \mathbb{R}^d$ and $g \in \partial\|x\|^2$,

$$\|y\|^2 \geq \|x\|^2 + \langle g, y - x\rangle + \frac{\lambda}{2}\|y - x\|^2.$$

**Assumption 2.** $W$ has diameter at most $D$: $\forall\, x, y \in W$, $\|x - y\| \leq D$.

**Assumption 3.** $\ell(x, z)$ is $G$-Lipschitz in $x$: $\forall\, x \in W, z \in Z$, $\|\nabla\ell(x, z)\|_* \leq G$.

**Assumption 4.** $\ell(x, z)$ is $H$-smooth in $x$: $\forall\, x, y \in W, z \in Z$, $\|\nabla\ell(x, z) - \nabla\ell(y, z)\|_* \leq H\|x - y\|$.

**Assumption 5.** $\mathbb{E}[\|\nabla\mathcal{L}(x) - \nabla\ell(x, z)\|_*^2] \leq \sigma_G^2$ for all $x, z$.

**Assumption 6.** $\mathbb{E}[\|[\nabla\mathcal{L}(x) - \nabla\mathcal{L}(y)] - [\nabla\ell(x, z) - \nabla\ell(y, z)]\|_*^2] \leq \sigma_H^2\|x - y\|^2$ for all $x, y, z$.

Note that if $\ell$ is $G$-Lipschitz and $H$-smooth in $x$, then so it $\mathcal{L}$, and Assumption 5 and 6 hold with $\sigma_G \leq 2G$ and $\sigma_H \leq 2H$. Moreover, notice that Assumption 2 - 6 depend on the norm $\|\cdot\|$.

**Differential Privacy** We now provide a formal definition of our privacy metric, differential privacy (DP). The definition hinges on the notion of *neighboring datasets*: datasets $Z = (z_1, \ldots, z_T)$ and $Z' = (z'_1, \ldots, z'_T)$ in $\mathcal{Z}^T$ are said to be neighbors if $|Z - Z'| \triangleq |\{t \mid z_t \neq z'_t\}| = 1$.

**Definition 1** (($\epsilon, \delta$)-DP (Dwork and Roth, 2014)). A randomized algorithm $M : \mathcal{Z}^T \to \mathbb{R}^d$ is $(\epsilon, \delta)$-differentially private for $\epsilon, \delta \geq 0$ if for any neighboring $Z, Z' \in \mathcal{Z}^T$ and $S \in \mathbb{R}^d$:

$$\mathcal{P}\{M(Z) \in S\} \leq \exp(\epsilon)\, \mathcal{P}\{M(Z') \in S\} + \delta$$

An alternative definition is *Rényi differential privacy* (RDP), which is a generalization of DP that allows us to compose mechanisms more easily and achieve tighter privacy bounds in certain cases.

**Definition 2** (($\alpha, \epsilon$)-RDP (Mironov, 2017)). A randomized mechanism $M : \mathcal{Z}^T \to \mathbb{R}^d$ is said to be $(\alpha, \epsilon)$-Rényi differentially private for $\alpha > 1, \epsilon \geq 0$ if for any neighboring datasets $Z, Z' \in \mathcal{Z}^T$:

$$D_\alpha(M(Z)\|M(Z')) \leq \epsilon$$

where $D_\alpha(P\|Q) \triangleq \frac{1}{\alpha - 1} \log \mathbb{E}_{x \sim Q}\left(\frac{P(x)}{Q(x)}\right)$.

RDP can be easily converted to the usual $(\epsilon, \delta)$-DP as follows (Mironov, 2017): if a randomized algorithm $M$ is $(\alpha, \epsilon)$-RDP, then it is also $(\epsilon + \frac{\log 1/\delta}{\alpha - 1}, \delta)$-DP for all $\delta \in (0, 1)$. In particular, if $M$ is $(\alpha, \alpha\rho^2/2)$-RDP for all $\alpha > 1$, then it is also $(2\rho\sqrt{\log(1/\delta)}, \delta)$-DP for all $\delta \geq \exp(-\rho^2)$.

Throughout the paper, we also frequently use the notion of sensitivity:

**Definition 3** (Sensitivity). The sensitivity of a function $f : \mathcal{Z}^T \to \mathbb{R}^d$ w.r.t. norm $\|\cdot\|$ is:

$$\Delta_f = \sup_{|Z - Z'|=1} \|f(Z) - f(Z')\|_*.$$

Almost all methods for ensuring differential privacy involve injecting noise whose scale increases with the sensitivity of the output. In other words, small sensitivity implies high privacy.

## 2 Diffentially Private Online-to-Batch

In this section, we present our main differentially private online-to-batch algorithm. To start, we need to define *online convex optimization* (OCO). OCO is a "game" in which for $T$ rounds, $t = 1, \ldots, T$, an algorithms predicts a parameter $w_t \in W$. It then receives a convex loss $\ell_t : W \to \mathbb{R}$ and pays the loss $\ell_t(w_t)$. The quality of the algorithm is measured by the regret w.r.t. a competitor $u$, defined as $\text{Regret}_T(u) = \sum_{t=1}^{T} \ell_t(w_t) - \ell_t(u)$.

Online-to-batch algorithms (Cesa-Bianchi et al., 2004) convert OCO algorithms (online learners) into stochastic convex optimization algorithms. In particular, for $\beta_1, \ldots, \beta_T > 0$, the anytime online-to-batch conversion (Cutkosky, 2019) defines the $t$-th loss as $\ell_t(w) = \langle \beta_t \nabla\ell(x_t, z_t), w \rangle$, where $x_t = \sum_{i=1}^{t} \frac{\beta_i w_i}{\beta_{1:t}}$.[2] Convergence of anytime online-to-batch builds on the following key result, and its proof is in Appendix A for completeness.

**Theorem 1** (Cutkosky (2019)). *For any sequence of $\beta_t > 0, g_t \in \mathbb{R}^d$, suppose an online learner predicts $w_t$ and receives $t$-th loss $\ell_t(w) = \langle g_t, w \rangle$. Define $x_t = \sum_{i=1}^{t} \frac{\beta_i w_i}{\beta_{1:t}}$ where $\beta_{1:t} = \sum_{i=1}^{t} \beta_i$. Then for any convex and differentiable $\mathcal{L}$,*

$$\beta_{1:T}(\mathcal{L}(x_T) - \mathcal{L}(u)) \leq \text{Regret}_T(u) + \sum_{t=1}^{T} \langle \beta_t \nabla\mathcal{L}(x_t) - g_t, w_t - u \rangle, \ \forall u \in \mathbb{R}^d.$$

A tighter bound is possible (Joulani et al., 2020), but the simpler expression above suffices for our analysis. As an immediate result, choosing $\beta_t = 1$ and $g_t = \beta_t \nabla\ell(x_t, z_t)$ satisfies $\mathbb{E}[g_t|x_t] = \beta_t \nabla\mathcal{L}(x_t)$, so $\mathbb{E}[\mathcal{L}(x_T) - \mathcal{L}(u)] \leq \mathbb{E}[\text{Regret}_T(u)]/T$. Therefore, any online learner with sub-linear regret guarantees convergence for the last iterate $x_T$. Notice that due to the formulation of the anytime online-to-batch, the iterate $x_t$ is *stable*. Consider the case where $\beta_t = 1$ for all t. Then, we have $\|x_t - x_{t-1}\| = \|w_t - x_{t-1}\|/t \leq D/t$ *regardless of what the online learner does*. This guarantee is

---

[2]Throughout this paper, we denote $\beta_{1:t} = \sum_{i=1}^{t} \beta_i$.

significantly stronger than the classical online-to-batch result of Cesa-Bianchi et al. (2004). We would like to take advantage of this stability to design our private online-to-batch algorithm. Intuitively, the algorithm has much lower sensitivity due to the stability of the iterates, which can be exploited to improve privacy.

Our goal will be to make the entire sequence $g_1, \ldots, g_T$ private, which, in turn, makes the entire algorithm private. To do so, we would like to add noise to the $g_t$'s while still having $g_t$ be a good estimate of $\nabla\mathcal{L}(x_t)$. In standard non-private online-to-batch, $g_t$ is usually defined as the stochastic gradient $\nabla\ell(x_t, z_t)$. However, directly adding noise to $\nabla\ell(x_t, z_t)$ is not a good idea because its sensitivity is $O(1)$ (more specifically the sensitivity is bounded by $2G$ by Lipschitzness). Consequently, we need to add noise to $g_t$ whose variance is of order $O(1/\epsilon)$ in order to achieve $\epsilon$-differential privacy.

Instead, we express $g_t = \sum_{i=1}^{t}\delta_i$, where $\delta_i = \beta_i\nabla\ell(x_i, z_i) - \beta_{i-1}\nabla\ell(x_{i-1}, z_i)$ and $\beta_0 \equiv 0$. Since we assume $\ell$ is smooth, if we set $\beta_i = 1$ then $\|\delta_i\|_* \leq H\|x_i - x_{i-1}\| \leq DH/i$, i.e., the sensitivity of $\delta_i$ is $O(1/i)$. As a result, we can privately estimate $g_t$ with error roughly $\tilde{O}(1/t\epsilon)$ using the *tree aggregation mechanism* (Dwork et al., 2010; Chan et al., 2011), an advanced technique that privately estimates running sums, such as our $\sum_{i=1}^{t}\delta_i$. Compared to directly adding noise to $\nabla\ell(x_t, z_t)$, this method adds less noise ($\tilde{O}(1/t\epsilon)$ compared to $O(1/\epsilon)$) and thus allows us to achieve the optimal rate.

On the other hand, after using these advanced aggregation techniques, $g_t$ is no longer an conditionally unbiased estimator of $\beta_t\nabla\mathcal{L}(x_t)$. More specifically, although it remains the case that $\mathbb{E}_{z_i}[\delta_i|z_1, \ldots, z_{i-1}] = \beta_i\nabla\mathcal{L}(x_i) - \beta_{i-1}\nabla\mathcal{L}(x_{i-1})$, it is not necessarily true that $\mathbb{E}[g_t|z_1, \ldots, z_{t-1}] \neq \beta_t\nabla\mathcal{L}(x_t)$. Unbiasedness plays a key role in standard convergence analyses, but we will need a much more delicate analysis.

Moreover, although we mostly discuss the $\beta_t = 1$ case above for intuition, our algorithm is analyzed using the general case $\beta_t = t^k$ for $k \geq 1$. The guiding principle for this formula is the sensitivity of $\delta_t$. For $k \geq 1$, the sensitivity of $\delta_t$ is of order $O(t^{k-1})$. For $k = 1$, this is a constant sensitivity, which is particularly intuitive for analysis. For $k = 0$ (i.e. the standard weighting in online-to-batch), the sensitivity is actually $O(1/t)$, which is much more complicated to analyze. In order to apply the tree aggregation easily, we want the sensitivity of $\delta_t$ to be polynomial in $t$, rather than the inverse polynomial $1/t$, so we define $\beta_t = t^k$ and ask $t \geq k$. Furthermore, in all cases except for the parameter-free case (Section 5), our results hold for all $k \geq 1$. In the parameter-free case, we choose $k = 3$ for algebraic reasons.

The pseudo-code is presented in Algorithm 1, which has linear time complexity. It is similar to anytime online-to-batch, while we replace $g_t$ with the more complicated definition and add noise $\gamma_t$ generated by the NOISE subroutine, which implements the tree aggregation. More specifically, given random noises $\{R_t\}$, NOISE($t$) returns $\sum_{i\in I_t} R_i$, where $I_t$ is the set of cumulative sums of the binary expansion of $t$. That is, for some $n \geq \lfloor\log_2(t)\rfloor + 1$, we define $\text{bin}(t) \in \{0,1\}^n$ by $t = \sum_{i=1}^{n}\text{bin}(t)[i]2^{n-i}$. Then, $I_t$ consists of all non-zero sums of the form $\sum_{k=1}^{i}\text{bin}(t)[i]2^{n-i}$. For examples, $7 = 4 + 2 + 1$, so $I_7 = \{4, 6, 7\}$; $8 = 8$, so $I_8 = \{8\}$.

---

**Algorithm 1** Differentially-Private Online-to-Batch
***
1: **Input:** OCO algorithm $\mathcal{A}$ with domain $W$, positive sequence $\{\beta_t\}$, distribution $\mathcal{D}$, dataset $Z$.
2: **Initialize:** Set $\beta_0 = 0, x_0 = 0$ and $g_0 = 0$. Set global variable $\mathcal{R} = \{\}$.
3: **for** $t = 1, \ldots, T$ **do**
4:      Get $w_t$ from $\mathcal{A}$ and compute $x_t = (\beta_{1:t-1}x_{t-1} + \beta_t w_t)/\beta_{1:t}$.
5:      Compute gradient difference $\delta_t = \beta_t\nabla\ell(x_t, z_t) - \beta_{t-1}\nabla\ell(x_{t-1}, z_t)$.
6:      Update $g_t = g_{t-1} + \delta_t$ and generate noise $\gamma_t = $ NOISE($t$).
7:      Send $\ell_t(w) = \langle g_t + \gamma_t, w\rangle$ to $\mathcal{A}$ as the $t$-th loss.

8: **function** NOISE($t$)
9:      **Initialize:** Set $k = 0, I_t = \{\}$, $\text{bin}(t)$ be the binary encoding of $t$, and $n = \lfloor\log_2 t\rfloor + 1$.
10:     **for** $i = 1, \ldots, n$ **do**
11:        If $\text{bin}(t)[i] = 1$, update $k = k + 2^{n-i}$ and $I_t = I_t \cup \{k\}$.
12:     Generate noise $\tilde{R}_t \sim \mathcal{D}$, compute $R_t = \sigma_t\tilde{R}_t$, and update $\mathcal{R} = \mathcal{R} \cup \{R_t\}$.
13:     **Return** $\gamma_t = \sum_{i\in I_t} R_i$.

---

**Convergence** Following Theorem 1 and Fenchel-Young's inequality, Algorithm 1 satisfies:

$$\beta_{1:T}(\mathcal{L}(x_T) - \mathcal{L}(x^*)) \leq \text{Regret}_T(x^*) + \sum_{t=1}^{T}\langle\beta_t\nabla\mathcal{L}(x_t) - g_t - \gamma_t, w_t - x^*\rangle \tag{2}$$

$$\leq \text{Regret}_T(x^*) + \sum_{t=1}^{T} D\|\beta_t\nabla\mathcal{L}(x_t) - g_t\|_* + \sum_{t=1}^{T} D\|\gamma_t\|_*. \tag{3}$$

(3) decomposes the suboptimality gap $\mathcal{L}(x_T) - \mathcal{L}(x^*)$ into three components: (i) regret $\text{Regret}_T(x^*)$, (ii) error associated with variance of $g_t$, measured by $\|\beta_t\nabla\mathcal{L}(x_t) - g_t\|_*$, and (iii) error from DP mechanism, measured by $\|\gamma_t\|_*$. To get a tight bound, we observe that $\beta_t\nabla\mathcal{L}(x_t) - g_t$ and $\gamma_t$ are sums of conditionally mean-zero random vectors (Lemma 15 in Appendix A). With a martingale bound in high dimension with general norm (Lemma 13), we can derive the following bounds. In particular, we show (Lemma 15) that if Assumption 1 - 6 hold and set $\beta_t = t^k$, then

$$\mathbb{E}[\|\beta_t\nabla\mathcal{L}(x_t) - g_t\|_*^2] \leq 4(k+1)^2(\sigma_G^2 + D^2\sigma_H^2)t^{2k-1}/\lambda, . \tag{4}$$

Moreover, we show (Lemma 16) that if $\mathbb{E}[R_t] = 0$ and $\mathbb{E}[\|R_t\|_*^2] \leq \bar{\sigma}_t^2$, then

$$\mathbb{E}[\|\gamma_t\|_*^2] \leq 2(\max_{i\leq t}\bar{\sigma}_i^2)\log_2(2t)/\lambda. \tag{5}$$

**Privacy** The next step is to determine how much noise is sufficient for Algorithm 1 to be RDP. We first make the following assumption on the distribution $\mathcal{D}$.

**Definition 4** (($V, \alpha$)-RDP distribution). *A distribution $\mathcal{D}$ on $\mathbb{R}^d$ is said to be a ($V, \alpha$)-RDP distribution on $\|\cdot\|$ if it satisfies that for $R \sim \mathcal{D}$ (i) $\mathbb{E}[R] = 0$, (ii) $\mathbb{E}[\|R\|_*^2] \leq V$, and (iii) for all $\rho > 0$ and $\mu, \mu' \in \mathbb{R}^d$, if $\sigma^2 \geq \|\mu - \mu'\|_*^2/\rho^2$, then $D_\alpha(\sigma R + \mu\|\sigma R + \mu') \leq \alpha\rho^2/2$.*[3]

**Remark 2.** A standard ($V, \alpha$)-RDP distribution on the $L_2$ norm is the multivariate normal distribution. Let $R \sim \mathcal{N}(0, I)$ then it is clear that $\mathbb{E}[R] = 0$ and $\mathbb{E}[\|R\|_2^2] = d$. For the third condition, we can show that for all $\mu, \mu'$ and $\alpha > 1$, $D_\alpha(\mathcal{N}(\mu, \sigma^2 I)\|\mathcal{N}(\mu', \sigma^2 I)) \leq \alpha\|\mu - \mu'\|_2^2/2\sigma^2$, which is further bounded by $\alpha\rho^2/2$ for all $\sigma^2 \geq \|\mu - \mu'\|_2^2/\rho^2$ (Lemma 18). Therefore, $\mathcal{N}(0, I)$ is a ($d, \alpha$)-RDP distribution for all $\alpha > 1$.

As its name suggests, adding noise sampled from RDP distribution is sufficient to make a deterministic algorithm RDP. Consider function $\hat{f}(Z) = f(Z) + \sigma R$, where $R \sim \mathcal{D}$ and $\mathcal{D}$ is a ($V, \alpha$)-RDP distribution on $\|\cdot\|$. Let $\Delta$ be the sensitivity of $f$ w.r.t $\|\cdot\|$. Set $\sigma^2 \geq \Delta^2/\rho^2$, then by Definition 4,

$$D_\alpha(\hat{f}(Z)\|\hat{f}(Z')) = D_\alpha(f(Z) + \sigma R\|f(Z') + \sigma R) \leq \alpha\rho^2/2.$$

Thus, $\hat{f}$ is ($\alpha, \alpha\rho^2/2$)-RDP. Moreover, we can compose RDP mechanisms via the tree aggregation mechanism. Specifically, we set $\beta_t = t^k$ and define the variance $\sigma_t^2$ in Algorithm 1 as follows:

$$\sigma_t^2 = \frac{4(k+1)^2}{\rho^2}(G + H\max_{i\in[t]}\|w_i - x_{i-1}\|)^2\log_2(2T)t^{2k-2}, \tag{6}$$

We assume $\|w_i - x_{i-1}\| \leq D$. Upon substituting (6) into (5), we get $\mathbb{E}[\|R_t\|_*^2] \leq \bar{\sigma}_t^2 \leq V\sigma_t^2$ and:

$$\mathbb{E}[\|\gamma_t\|_*^2] \leq \frac{8(k+1)^2 V(G + DH)^2}{\lambda\rho^2}\log_2^2(2T)t^{2k-2}. \tag{7}$$

The following theorem shows that Algorithm 1 is Rényi differentially private if we define $\sigma_t^2$ as in (6). Its proof is presented in Appendix B. Note that the privacy guarantee does *not* require i.i.d. $Z$.

**Theorem 3.** *Suppose $\|\cdot\|^2$ is $\lambda$-strongly convex, $W$ is bounded by $D$, $\ell$ is $G$-Lipschitz and $H$-smooth, and $\mathcal{D}$ is a ($V, \alpha$)-RDP distribution. If $\beta_t = t^k$ and $\sigma_t^2$ is as defined in (6), then Algorithm 1 is ($\alpha, \alpha\rho^2/2$)-DP for all datasets $Z$.*

---

[3]Here we slightly abuse the notation. For random vectors $X, Y$, $D_\alpha(X\|Y)$ denotes the Rényi divergence of the underlying distributions of $X$ and $Y$.

**Main Result.** Now we can combine all the previous results to prove the privacy and convergence guarantee of our algorithm.

**Theorem 4.** *Suppose Assumption 1 - 6 hold, and $\mathcal{D}$ is a $(V, \alpha)$-RDP distribution. If we set $\beta_t = t^k$ and define $\sigma_t^2$ as in (6), then Algorithm 1 is $(\alpha, \alpha\rho^2/2)$-RDP and $\mathbb{E}[\mathcal{L}(x_T) - \mathcal{L}(x^*)]$ is bounded by:*

$$\frac{(k+1)\,\mathbb{E}[\mathrm{Regret}_T(x^*)]}{T^{k+1}} + \frac{2(k+1)^2 D}{\sqrt{\lambda}}\left(\frac{\sigma_G + D\sigma_H}{\sqrt{T}} + \frac{\sqrt{2V}(G + DH)\log_2(2T)}{\rho T}\right).$$

*Moreover, recall that the online learner receives $t$-th loss $\ell_t(w) = \langle g_t + \gamma_t, w\rangle$. It holds that*

$$\mathbb{E}[\|g_t + \gamma_t\|_*^2] \leq 3t^{2k}\left(G^2 + \frac{4(k+1)^2}{\lambda}\left(\frac{(\sigma_G + D\sigma_H)^2}{t} + \frac{2V(G + DH)^2\log_2^2(2T)}{\rho^2 t^2}\right)\right).$$

**Remark 5.** As an example, let's consider the Gaussian distribution $\mathcal{N}(0, I)$ on the 2-norm, which is a $(d, \alpha)$-RDP distribution for all $\alpha > 1$ (Remark 2). For many popular online learners (OSD, OMD, FTRL), if $\mathbb{E}[\|\nabla\ell_t(w_t)\|_*^2] \leq \hat{G}^2$ for all $t$, then $\mathbb{E}[\mathrm{Regret}_T(x^*)] \leq O(D\hat{G}\sqrt{T})$. Hence, Theorem 4 with $\mathcal{D} = \mathcal{N}(0, I)$ implies that

$$\mathbb{E}[\mathcal{L}(x_T) - \mathcal{L}(x^*)] = O\left(\frac{D(G + D\sigma_H)}{\sqrt{T}} + \frac{\sqrt{d}D(G + DH)\log T}{\rho T}\right).$$

This bound is of $\tilde{O}(1/\sqrt{T} + \sqrt{d}/\rho T)$, which can be translated to an equivalent $(\epsilon, \delta)-$DP bound of $\tilde{O}(1/\sqrt{T} + \sqrt{d\log(1/\delta)}/\epsilon T)$. This bound matches the optimal rate for private stochastic optimization with convex and smooth losses (Bassily et al. (2019)).

*Proof of Theorem 4.* From Eq. (3), we have:

$$\mathbb{E}[\mathcal{L}(x_T) - \mathcal{L}(x^*)] \leq \frac{1}{\beta_{1:T}}\mathbb{E}\left[\mathrm{Regret}_T(x^*) + D\sum_{t=1}^{T}(\|\beta_t\nabla\mathcal{L}(x_t) - g_t\|_* + \|\gamma_t\|_*)\right]$$

Recall the bounds of $\mathbb{E}[\|\beta_t\nabla\mathcal{L}(x_t) - g_t\|_*^2]$ and $\mathbb{E}[\|\gamma_t\|_*^2]$ in (4) and (7). By Jensen's inequality, $\mathbb{E}[\|X\|_*] \leq \sqrt{\mathbb{E}[\|X\|_*^2]}$. Moreover, since $\beta_t = t^k$ and $k \geq 1$, it holds that $\beta_{1:T} \geq T^{k+1}/k + 1$, so:

$$\leq \frac{\mathbb{E}[\mathrm{Regret}]}{\beta_{1:T}} + \frac{D}{\beta_{1:T}}\sum_{t=1}^{T}\frac{2(k+1)(\sigma_G + D\sigma_H)t^{k-\frac{1}{2}}}{\sqrt{\lambda}} + \frac{\sqrt{8V}(k+1)(G + DH)\log_2(2T)t^{k-1}}{\sqrt{\lambda}\rho}$$

$$\leq \frac{(k+1)\,\mathbb{E}[\mathrm{Regret}_T(x^*)]}{T^{k+1}} + \frac{2(k+1)^2 D}{\sqrt{\lambda}}\left(\frac{\sigma_G + D\sigma_H}{\sqrt{T}} + \frac{\sqrt{2V}(G + DH)\log_2(2T)}{\rho T}\right).$$

For the second part of the theorem,

$$\mathbb{E}[\|g_t + \gamma_t\|_*^2] \leq 3\,\mathbb{E}[\|\beta_t\nabla\mathcal{L}(x_t)\|_*^2 + \|g_t - \beta_t\nabla\mathcal{L}(x_t)\|_*^2 + \|\gamma_t\|_*^2]$$

We bound the first term by Lipschitzness, the second by (4), and the third by (7)

$$\leq 3t^{2k}G^2 + \frac{12(k+1)^2(\sigma_G^2 + D^2\sigma_H^2)t^{2k-1}}{\lambda} + \frac{24V(k+1)^2(G + DH)^2\log_2^2(2T)t^{2k-2}}{\lambda\rho^2}$$

$$= 3t^{2k}\left(G^2 + \frac{4(k+1)^2}{\lambda}\left(\frac{(\sigma_G + D\sigma_H)^2}{t} + \frac{2V(G + DH)^2\log_2^2(2T)}{\rho^2 t^2}\right)\right).$$

$\square$

## 3 The Optimistic Case

In this section, we show that choosing an optimistic online learner (Chiang et al., 2012; Rakhlin and Sridharan, 2013; Steinhardt and Liang, 2014) will accelerate our DP online-to-batch algorithm. Optimistic algorithms are provided with additional "hints" in the form of $\hat{\ell}_t(w) = \langle \hat{g}_t, w\rangle$ as an

approximation of the true loss $\ell_t(w) = \langle g_t, w \rangle$, and they can incorporate $\hat{\ell}_t$ to decide $w_t$. The regret of an optimistic algorithm depends on the quality of hints: if $\hat{g}_t \approx g_t$, then it achieves a low regret. Formally, in this paper, we say an online learning algorithm is *optimistic w.r.t. norm* $\|\cdot\|$ if its regret is the following:

$$\text{Regret}_T(x^*) \le O\left( D\sqrt{\sum_{t=1}^{T} \|\hat{g}_t - g_t\|_*^2} \right), \tag{8}$$

where $D$ denotes the diameter of the learner's domain.

A common choice of the hint $\hat{g}_t$ is $g_{t-1}$, the gradient in the last round since intuitively, one could expect $g_{t-1} \approx g_t$ when the loss functions are smooth. In this section, we also follow this choice. Recall that in Algorithm 1, the online learner receives $t$-th loss $\ell_t(w) = \langle g_t + \gamma_t, w \rangle$, where $g_t = \sum_{i=1}^{t} \delta_i$ is the sum of gradient differences, and $\gamma_t$ is some noise. Therefore, we define the $t$-th hint as $\hat{g}_t = g_{t-1} + \gamma_{t-1}$.

**Theorem 6.** *Suppose Assumption 1 - 4 hold, and $\mathcal{D}$ is a $(V, \alpha)$-RDP distribution. Set $\beta_t = t^k$ and $\sigma_t^2$ as defined in (6), If the online learner is optimistic (satisfying (8)) with $t$-th gradient $\bar{g}_t = g_t + \gamma_t$ and $t$-th hint $\hat{g}_t = \bar{g}_{t-1}$, then*

$$\frac{\mathbb{E}[\text{Regret}_T(x^*)]}{\beta_{1:T}} \le O\left( D(G + DH)\left(1 + \frac{\sqrt{V}\log T}{\sqrt{\lambda}\rho}\right)\frac{1}{T^{3/2}} \right).$$

The proof is in Appendix D. As an immediate corollary, if we further assume Assumption 5 and 6, then Theorem 4 applies. Together with this theorem, they imply that optimistic learners achieve the following convergence rate that is adaptive to the variance:

$$\mathbb{E}[\mathcal{L}(x_T) - \mathcal{L}(x^*)] = O\left( \frac{D(\sigma_G + D\sigma_H)}{\sqrt{T}} + \frac{\sqrt{d}D(G + DH)\log T}{\rho T} \right).$$

Compared to the bound in the non-optimistic case (Remark 5), this bound has $\sigma_G$ instead of $G$ in the first term. Thus, when the gradient $\nabla\ell(x_t, z_t)$ has low variance, i.e., $\sigma_G \ll G$, the optimistic bound outperforms the standard bound.

## 4 The Strongly Convex Case

In this section, we prove that in the case of strong convexity, our algorithm can be improved by regularizing the loss of online learner. If $\mathcal{L}$ is strongly convex, then we can prove a similar result to Theorem 1 (the proof is in Appendix E).

**Lemma 7.** *Suppose $\mathcal{L}$ is $\mu$-strongly convex w.r.t. $\|\cdot\|$. If we replace $\ell_t(w) = \langle g_t + \gamma_t, w \rangle$ with $\bar{\ell}_t(w) = \ell_t(w) + \frac{\beta_t \mu}{4}\|w - x_t\|^2$ in Algorithm 1, and denote the associated regret by $\overline{\text{Regret}}_T$, then*

$$\beta_{1:T}(\mathcal{L}(x_T) - \mathcal{L}(x^*)) \le \overline{\text{Regret}}_T(x^*) + \sum_{t=1}^{T}\langle \beta_t\nabla\mathcal{L}(x_t) - g_t - \gamma_t, w_t - x^*\rangle - \frac{\beta_t\mu}{8}\|w_t - x^*\|^2$$

$$\le \overline{\text{Regret}}_T(x^*) + \sum_{t=1}^{T}\frac{2\|\beta_t\nabla\mathcal{L}(x_t) - g_t - \gamma_t\|_*^2}{\beta_t\mu}.$$

Compared to Lemma 1 and Equation 3, in the strongly convex case, there is an additional term $-\frac{\beta_t\mu}{8}\|w_t - x^*\|^2$, which allows the improved convergence rate:

**Theorem 8.** *Suppose Assumption 1 - 6 hold, and $\mathcal{D}$ is a $(V, \alpha)$-RDP distribution. Also suppose $\mathcal{L}$ is $\mu$-strongly convex. Set $\beta_t = t^k$ and $\sigma_t^2$ as defined in (6). Then $\mathbb{E}[\mathcal{L}(x_T) - \mathcal{L}(x^*)]$ is bounded by:*

$$\frac{(k+1)\mathbb{E}[\overline{\text{Regret}}_T(x^*)]}{T^{k+1}} + \frac{16(k+1)^3}{\lambda\mu}\left( \frac{(\sigma_G + D\sigma_H)^2}{T} + \frac{2V(G + DH)^2\log_2^2(2T)}{\rho^2 T^2} \right).$$

*Moreover, for all $\bar{g}_t \in \partial\bar{\ell}_t(w_t)$,*

$$\mathbb{E}[\|\bar{g}_t\|_*^2] \le t^{2k}\left( 4G^2 + \mu^2 D^2 + \frac{16(k+1)^2}{\lambda}\left( \frac{(\sigma_G + D\sigma_H)^2}{t} + \frac{2V(G + DH)^2\log_2^2(2T)}{\rho^2 t^2} \right) \right).$$

**Remark 9.** As an example, again consider the case $\mathcal{D} = \mathcal{N}(0, I)$ with $L_2$ norm. Online subgradient descent (OSD) with appropriate learning rate on $\mu_t$-strongly convex losses $\ell_t$ achieves:

$$\text{Regret}_T(u) \leq \sum_{t=1}^{T} \frac{\|g_t\|_2^2}{2 \sum_{i=1}^{t} \mu_i},$$

where $g_t \in \partial \ell_t(w_t)$. In our case, $\|\cdot\|_2^2$ is 2-strongly convex and the regularized loss $\bar{\ell}_t$ is $\frac{\beta_t \mu}{2}$-strongly convex, i.e. $\mu_t = \mu t^k / 2$ and $\sum_{i=1}^{t} \mu_i = O(\mu t^{k+1})$. Therefore, by Theorem 8,

$$\mathbb{E}[\overline{\text{Regret}}_T(x^*)] \leq O\left( \frac{T^k}{\mu} \left( G^2 + \mu^2 D^2 + \frac{(\sigma_G + D\sigma_H)^2}{T} + \frac{d(G + DH)^2 \log^2 T}{\rho^2 T^2} \right) \right).$$

Consequently,

$$\mathbb{E}[\mathcal{L}(x_T) - \mathcal{L}(x^*)] \leq O\left( \frac{(G + \mu D + D\sigma_H)^2}{\mu T} + \frac{d(G + DH)^2 \log^2 T}{\mu \rho^2 T^2} \right).$$

This again matches the optimal private convergence rates.

*Proof of Theorem 8.* We have already bounded $\mathbb{E}[\|\beta_t \nabla \mathcal{L}(x_t) - g_t\|_*^2]$ and $\mathbb{E}[\|\gamma_t\|_*^2]$ in (4) and (7) respectively, so

$$\mathbb{E}[\|\beta_t \nabla \mathcal{L}(x_t) - g_t - \gamma_t\|_*^2] \leq 2 \mathbb{E}[\|\beta_t \nabla \mathcal{L}(x_t) - g_t\|_*^2 + \|\gamma_t\|_*^2]$$
$$\leq \frac{8(k+1)^2 t^{2k-1}}{\lambda} \left( (\sigma_G + D\sigma_H)^2 + \frac{2V(G + DH)^2 \log_2^2(2T)}{\rho^2 t} \right).$$

Upon substituting this into Lemma 7 and replace $\beta_t = t^k$, we get:

$$\beta_{1:T} \, \mathbb{E}[\mathcal{L}(x_T) - \mathcal{L}(x^*)]$$
$$\leq \mathbb{E}[\overline{\text{Regret}}_T(x^*)] + \sum_{t=1}^{T} \frac{16(k+1)^2 t^{k-1}}{\lambda \mu} \left( (\sigma_G + D\sigma_H)^2 + \frac{2V(G + DH)^2 \log_2^2(2T)}{\rho^2 t} \right)$$
$$\leq \mathbb{E}[\overline{\text{Regret}}_T(x^*)] + \frac{16(k+1)^2}{\lambda \mu} \left( (\sigma_G + D\sigma_H)^2 T^k + \frac{2V(G + DH)^2 \log_2^2(2T) T^{k-1}}{\rho^2} \right).$$

Dividing both sides by $\beta_{1:T} \geq T^{k+1}/(k+1)$ proves the first part of the theorem.

For the second part, recall that $\bar{\ell}_t(w) = \langle g_t + \gamma_t, w \rangle + \frac{\beta_t \mu}{4} \|w - x_t\|^2$. Therefore, for all $\bar{g}_t \in \partial \bar{\ell}_t(w)$, $\bar{g}_t = g_t + \gamma_t + \frac{\beta_t \mu}{4} v$, where $v \in \partial \|w_t - x_t\|^2$. We follow the same argument in Theorem 4:

$$\mathbb{E}[\|\bar{g}_t\|_*^2] \leq 4 \mathbb{E}[\|\beta_t \nabla \mathcal{L}(x_t)\|_*^2 + \|\beta_t \nabla \mathcal{L}(x_t) - g_t\|_*^2 + \|\gamma_t\|_*^2 + \|\tfrac{\beta_t \mu}{4} v\|_*^2]$$

Here $v \in \partial \|w_t - x_t\|^2$. We bound the first term by Lipschitz, the second by (4), and the third by (7). Moreover, by chain rule (Proposition 21), we can show that $\|v\|_* \leq 2D$, so:

$$\leq 4t^{2k} G^2 + \frac{16(k+1)^2 (\sigma_G + D\sigma_H)^2 t^{2k-1}}{\lambda}$$
$$+ \frac{32V(k+1)^2 (G + DH)^2 t^{2k-2} \log_2^2(2T)}{\lambda \rho^2} + \mu^2 D^2 t^{2k}$$
$$\leq t^{2k} \left( 4G^2 + \mu^2 D^2 + \frac{16(k+1)^2}{\lambda} \left( \frac{(\sigma_G + D\sigma_H)^2}{t} + \frac{2V(G + DH)^2 \log_2^2(2T)}{\rho^2 t^2} \right) \right).$$

$\square$

## 5 Parameter-free Algorithm

In this section, we apply Algorithm 1 with a "parameter-free" online learner. These are algorithms that guarantee $\text{Regret}_T(u) \leq \tilde{O}(\|u\| \sqrt{T})$ for all competitors $u$ simultaneously (Orabona and Pál,

2016; Cutkosky and Orabona, 2018; Mhammedi and Koolen, 2020). By shifting coordinates, it is possible to obtain $\mathrm{Regret}_T(u) \leq \tilde{O}(\|u - x_0\|\sqrt{T})$ for any pre-specified point $x_0$. Thus, if $x_0 \approx u$ is some good initialization, perhaps generated by pretraining, this bound yields significantly smaller regret than if we had used the worst-case diameter bound $\|u - x_0\| \leq D$.

In order to obtain this refined bound with privacy, we need to make a small modification to our conversion. For simplicity, we focus on Euclidean space with 2-norm, and we assume the distribution $\mathcal{D}$ is in addition sub-Gaussian, i.e., for $R \sim \mathcal{D}$,

$$\mathcal{P}\{\sup_{\|a\|_2=1} \langle R, a \rangle \geq \epsilon\} \leq \exp\left(-\frac{\epsilon^2}{2\sigma^2}\right).$$

In general, the proof extends to any Banach space and any distribution $\mathcal{D}$ that concentrates on it.

In previous analysis (Equation (3)), we roughly bound $\|w_t - x^*\| \leq D$. However, in this section, we come up with a finer high probability bound that maintains a dependence on $\|w_t\|, \|x^*\|$. We then replace the loss $\ell_t(w)$ in Algorithm 1 with a regularized loss $\ell_t(w) + \xi_t\|w\|_2 + \nu_t\|w\|_2^2$, and we show that the new algorithm with regularized loss can achieve a parameter-free bound. The complete proof is presented in Appendix F.

**Theorem 10.** *Suppose w.r.t. 2-norm, $W$ is bounded by $D$ and $\ell$ is $G$-Lipschitz and $H$-smooth. Suppose $\mathcal{D}$ is $(V, \alpha)$-RDP distribution and is $\sigma_{\mathcal{D}}$-sub-Gaussian. If we set $\beta_t = t^3$ (i.e. $k = 3$) and set $\sigma_t^2$ as defined in (6), then with probability at least $1 - \delta$,*

$$\mathcal{L}(x_T) - \mathcal{L}(x^*) \leq \frac{4}{T^4}\left(\mathrm{Regret}_T(x^*) + \sum_{t=1}^{T} \xi_t(\|w_t\|_2 + \|x^*\|_2) + \nu_t(\|w_t\|_2^2 + \|x^*\|_2^2)\right).$$

*where $C$ is a universal constant, $A = 8\sqrt{2}C^2, A' = 8\sqrt{d}\sigma_{\mathcal{D}}C^2, \kappa = 1 + DH/G$, and*

$$\xi_t = AG\Phi t^{5/2} + A'(G + DH)\frac{\Phi \log_2(2T)t^2}{\rho}, \ \nu_t = 28AH\Phi t^{5/2}, \ \Phi = \sqrt{\log\frac{20dT\log(2\kappa T)}{\delta}}.$$

**Theorem 11.** *Following the assumptions and notations in Theorem 10, if we replace $\ell_t(w)$ in Algorithm 1 with regularized loss $\bar{\ell}_t(w) = \ell_t(w) + \xi_t\|w\|_2 + \nu_t\|w\|_2^2$ and denote the associated regret as $\overline{\mathrm{Regret}}_t$, then with probability at least $1 - \delta$, $\mathcal{L}(x_T) - \mathcal{L}(x^*)$ is bounded by:*

$$\frac{4\overline{\mathrm{Regret}}_T(x^*)}{T^4} + \frac{8A\|x^*\|(G + 28\|x^*\|H)\Phi}{\sqrt{T}} + \frac{8A'\|x^*\|(G + DH)\Phi \log_2(2T)}{\rho T}.$$

*Moreover, with probability at least $1 - \delta$, for all $t$ and for all $w \in W, \bar{g}_t \in \partial\bar{\ell}_t(w)$,*

$$\|\bar{g}_t\|_2 \leq Gt^3 + A(2G + 57DH)\Phi t^{5/2} + 2A'(G + DH)\frac{\Phi \log_2(2T)t^2}{\rho}.$$

**Remark 12.** *If $\|\bar{g}_t\|_2 \leq \hat{G}$ for all $t$, parameter-free algorithms achieve regret bound $\mathrm{Regret}_T(u) = \tilde{O}(\|u\|_2\hat{G}\sqrt{T})$. Therefore, with $\mathcal{D} = \mathcal{N}(0, I)$, $\mathcal{D}$ is 1-sub-Gaussian and $A' = O(\sqrt{d})$, so Theorem 11 implies that with probability $1 - \delta$,*

$$\frac{\overline{\mathrm{Regret}}_T(x^*)}{T^4} = \tilde{O}\left(\frac{\|x^*\|_2 G}{\sqrt{T}} + \frac{\|x^*\|_2(G + DH)}{T} + \frac{\|x^*\|_2\sqrt{d}(G + DH)}{\rho T^{3/2}}\right).$$

*Consequently,*

$$\mathcal{L}(x_T) - \mathcal{L}(x^*) \leq \tilde{O}\left(\frac{\|x^*\|_2(G + DH)}{\sqrt{T}} + \frac{\|x^*\|_2\sqrt{d}(G + DH)}{\rho T}\right).$$

*Proof of Theorem 11.* The regularized regret satisfies

$$\overline{\mathrm{Regret}}_T(x^*) = \mathrm{Regret}_T(x^*) + \sum_{t=1}^{T} \xi_t(\|w_t\|_2 - \|x^*\|_2) + \nu_t(\|w_t\|_2^2 - \|x^*\|_2^2).$$

Hence, upon substituting this equation into Theorem 10, we get: with probability at least $1 - \delta$,

$$\mathcal{L}(x_T) - \mathcal{L}(x^*) \leq \frac{4\overline{\mathrm{Regret}}_T(x^*)}{T^4} + \frac{8}{T^4}\sum_{t=1}^{T}\xi_t\|x^*\|_2 + \nu_t\|x^*\|_2^2$$

$$\leq \frac{4\mathrm{Regret}_T(x^*)}{T^4} + \frac{8(AG\Phi\|x^*\|_2 + 28AH\Phi\|x^*\|_2^2)}{\sqrt{T}} + \frac{8A'(G + DH)\Phi\log_2(2T)\|x^*\|}{\rho T}.$$

The second inequality is from $\sum_{t=1}^{T}\xi_t \leq T\xi_T$ because $\xi_t$ is increasing with $t$ (so is $\nu_t$).

For the second part of the theorem, for each fixed $t$ and for all $\bar{g}_t \in \bar{\ell}_t(w_t)$, $\bar{g}_t = g_t + \gamma_t + \xi_t u + 2\nu_t w_t$, where $u \in \partial\|w_t\|_2$ and thus $\|u\|_2 \leq 1$. Therefore,

$$\|\bar{g}_t\|_2 \leq \|g_t - \beta_t\nabla\mathcal{L}(x_t)\|_2 + \|\beta_t\nabla\mathcal{L}(x_t)\|_2 + \|\gamma_t\|_2 + \|\xi_t u\|_2 + \|2\nu_t w\|_2$$

By Lipschitzness, $\|\beta_t\nabla\mathcal{L}(x_t)\|_2 \leq Gt^3$. Since $W$ is bounded, $\|2\nu_t w_t\|_2 \leq 2D\nu_t$. Also, $\|\xi_t u\|_2 \leq \xi_t$. Moreover, we can prove (in Eq. 14 and 17) that for each $t$, with probability at least $1 - \delta/2T$,

$$\|\beta_t\nabla\mathcal{L}(x_t) - g_t\|_2 \leq 8C^2\Phi\sqrt{\sum_{i=1}^{t}i^4(G + H\|w_i - x_{i-1}\|_2)^2} \leq A\Phi(G + DH)t^{5/2},$$

$$\|\gamma_t\|_2 \leq \frac{A'}{\rho}(G + DH)\Phi\log_2(2T)t^2.$$

Upon taking the union bound for all $t$ and the definition of $\xi_t, \nu_t$, we get the desired bound. $\qquad\square$

## 6   Conclusion

We have presented a new online-to-batch conversion that produces private stochastic optimization algorithms on smooth losses. Online algorithms achieving the optimal $O(\sqrt{T})$ regret automatically achieve the optimal $\tilde{O}(1/\sqrt{T} + \sqrt{d}/\epsilon T)$ convergence rate. Combining this technique with the literature on online learning can yield new private optimization algorithms.

**Limitations:** Our algorithm requires smoothness, and unlike some other bounds, we cannot tolerate large $H$. In the worst case when $H = \sqrt{T}$ and $\sigma_H = H$, our standard bound in Remark 5 becomes $O(1)$. In other words, we need to assume $H = o(\sqrt{T})$ to ensure a non-trivial bound. Removing this restriction would significantly improve the generality of the procedure,

The dependence on $H$ comes from the sensitivity of $\delta_t$ (Lemma 15), where we apply smoothness to bound $\|\beta_t(\nabla\ell(x_t, z_t) - \nabla\ell(x_{t-1}, z_t))\| \leq \beta_t H\|x_t - x_{t-1}\|$ and use the stability of $x_t$ to further bound $\|x_t - x_{t-1}\| \leq D\beta_t/\beta_{1:t}$, which are necessary steps in order to bound the sensitivity of $\delta_t$ by $O(t^{k-1})$. Hence, it's not clear how to remove the smoothness assumption.

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
