\sum_{t=1}^{T} \beta_t \langle \nabla \mathcal{L}(x_t), x_t - w_t + w_t - u \rangle$$

$$= \sum_{t=1}^{T} \langle \nabla \mathcal{L}(x_t), \beta_t(x_t - w_t) \rangle + \langle \beta_t \nabla \mathcal{L}(x_t) - g_t + g_t, w_t - u \rangle.$$

By construction of $x_t$, it holds that $\beta_t(x_t - w_t) = \beta_{1:t-1}(x_{t-1} - x_t)$. By convexity,

$$\langle \nabla \mathcal{L}(x_t), \beta_{1:t-1}(x_{t-1} - x_t) \rangle \leq \beta_{1:t-1} \left( \mathcal{L}(x_{t-1}) - \mathcal{L}(x_t) \right).$$

Next, we move $\sum \beta_t \mathcal{L}(x_t)$ to the right, giving:

$$-\beta_{1:T}\mathcal{L}(u) \leq \sum_{t=1}^{T} \left( \beta_{1:t-1}\mathcal{L}(x_{t-1}) - \beta_{1:t}\mathcal{L}(x_t) + \langle \beta_t \nabla \mathcal{L}(x_t) - g_t + g_t, w_t - u \rangle \right)$$

$$= -\beta_{1:T}\mathcal{L}(x_T) + \text{Regret}_T(u) + \sum_{t=1}^{T} \langle \beta_t \nabla \mathcal{L}(x_T) - g_t, w_t - u \rangle,$$

where the equality follows from (i) the telescopic sum of $\beta_{1:t-1}\nabla \mathcal{L}(x_{t-1}) + \beta_{1:t}\nabla \mathcal{L}(x_t)$ and (ii) the definition of regret that $\text{Regret}_T(u) = \sum_{t=1}^{T} \langle g_t, w_t - u \rangle$. $\square$

**Lemma 13.** *Suppose $\| \cdot \|^2$ is $\lambda$-strongly convex w.r.t. $\| \cdot \|$, and let $\{X_t\}$ be a sequence of random vectors such that (i) $\mathbb{E}[\|X_t\|_*] < \infty$ and (ii) $\mathbb{E}[X_{t+1} \mid X_{1:t}] = 0$ for all $t$. Then,*

$$\mathbb{E}\left[ \left\| \sum_{t=1}^{T} X_t \right\|_*^2 \right] \leq \frac{2}{\lambda} \sum_{t=1}^{T} \mathbb{E}[\|X_t\|_*^2].$$

*Proof.* We use the regret approach to prove this statement. For simplicity, denote $M_T = \sum_{t=1}^{T} X_t$. Consider an online learner which receives $\ell_t(x) = \langle X_t, x \rangle$ as $t$-th loss and updates $w_{t+1}$. Then by definition of regret, for any $u$,

$$-\langle M_T, u \rangle \leq \text{Regret}_T(u) - \sum_{t=1}^{T} \langle X_t, w_t \rangle.$$

Since $w_t$ only depends on $X_{1:t-1}$ but *not* on $X_t$, $w_t$ is constant given $X_{1:t-1}$. Therefore,

$$\mathbb{E}[\langle X_t, w_t \rangle] = \mathop{\mathbb{E}}_{X_{1:t-1}} \mathop{\mathbb{E}}_{X_t} [\langle X_t, w_t \rangle | X_{1:t-1}]$$

$$= \mathop{\mathbb{E}}_{X_{1:t-1}} \left[ \langle \mathop{\mathbb{E}}_{X_t}[X_t | X_{1:t-1}], w_t \rangle \right] = 0,$$

where the second equality follows from the assumption that $\mathbb{E}[X_t | X_{1:t-1}] = 0$. Therefore,

$$\mathbb{E}[\langle M_T, -u \rangle] \leq \mathbb{E}[\text{Regret}_T(u)].$$

Recall the definition of the dual norm that $\|M_T\|_* = \sup_{\|x\|=1} \langle M_T, x \rangle$. Therefore, if we define $u^* = \|M_T\|_* \text{argmax}_{\|u\|=1} \langle M_T, -u \rangle$, then it holds that

$$\langle M_T, -u^* \rangle = \|M_T\|_*^2 \quad and \quad \|u^*\| = \|M_T\|_*.$$

Let the follow-the-regularized-leader (FTRL) algorithm be the online learner. Orabona (2019) proved that for any regularizer $\psi$ that is $\lambda$-strongly convex w.r.t. $\|\cdot\|$, FTRL achieves the following regret:

$$\text{Regret}_T(u) \leq \frac{\psi(u)}{\eta} + \frac{\eta}{2\lambda}\sum_{t=1}^{T}\|X_t\|_*^2.$$

Since we assume $\|\cdot\|^2$ is $\lambda$-strongly convex w.r.t. $\|\cdot\|$, we can define $\psi(x) = \|x\|^2$ and get:

$$\mathbb{E}[\langle M_T, -u^*\rangle] = \mathbb{E}[\|M_T\|_*^2] \leq \mathbb{E}[\text{Regret}_T(u^*)] \leq \mathbb{E}\left[\frac{\|M_T\|_*^2}{\eta} + \frac{\eta}{2\lambda}\sum_{t=1}^{T}\|X_t\|_*^2\right].$$

Equivalently, upon moving terms around we have:

$$\mathbb{E}[\|M_T\|_*^2] \leq \mathbb{E}\left[\frac{\eta^2}{2\lambda(\eta-1)}\sum_{t=1}^{T}\|X_t\|_*^2\right] \leq \mathbb{E}\left[\frac{2}{\lambda}\sum_{t=1}^{T}\|X_t\|_*^2\right].$$

The second inequality holds because $\inf_\eta \frac{\eta^2}{\eta-1} = 4$ when $\eta = 2$. $\qquad\square$

In this paper, we will always set $\beta_t = t^k$ for some $k \geq 1$. The following proposition gives relevant bounds for $\beta_t - \beta_{t-1}$ and $\beta_t^2/\beta_{1:t}$.

**Proposition 14.** *If $\beta_t = t^k$, then (i) $\beta_t - \beta_{t-1} \leq kt^{k-1}$ and (ii) $\beta_t^2/\beta_{1:t} \leq (k+1)t^{k-1}$.*

*Proof.* For the first part, by mean value theorem, there exists some $\tau \in [t-1, t]$ such that

$$\beta_t - \beta_{t-1} = t^k - (t-1)^k = k\tau^{k-1} \leq kt^{k-1}.$$

For the second part, for any increasing function $f$, it holds that $\sum_{i=1}^{t} f(i) \geq \int_0^t f(x)\,dx$, so

$$\beta_{1:t} = \sum_{i=1}^{t} i^k \geq \int_0^t x^k\,dx = \frac{t^{k+1}}{k+1}.$$

Hence, $\beta_t^2/\beta_{1:t} \leq (k+1)t^{k-1}$. $\qquad\square$

**Lemma 15.** *Suppose $\|\cdot\|^2$ is $\lambda$-strongly convex, $W$ is bounded by $D$, and $\ell$ is $G$-Lipschitz and $H$-smooth. If we set $\beta_t = t^k$, then*

$$\|\delta_t\|_* \leq (k+1)(G + H\|w_t - x_{t-1}\|)t^{k-1}.$$

*If we further assume Assumption 5 and 6, then $\beta_t\nabla\mathcal{L}(x_t) - g_t = \sum_{i=1}^{t} X_i$ such that:*

(i) $X_i = [\beta_i\nabla\mathcal{L}(x_i) - \beta_{i-1}\nabla\mathcal{L}(x_{i-1})] - [\beta_i\nabla\ell(x_i, z_i) - \beta_{i-1}\nabla\ell(x_{i-1}, z_i)]$,

(ii) $\mathbb{E}[X_i|z_{1:i-1}] = 0$, *and* (iii) $\mathbb{E}[\|X_i\|_*^2] \leq 2(k+1)^2(\sigma_G^2 + D^2\sigma_H^2)i^{2k-2}$.

*Consequently,*

$$\mathbb{E}[\|\beta_t\nabla\mathcal{L}(x_t) - g_t\|_*^2] \leq 4(k+1)^2(\sigma_G^2 + D^2\sigma_H^2)t^{2k-1}/\lambda.$$

*Proof.* For the first part, note that

$$\begin{aligned}\delta_t &= \beta_t\nabla\ell(x_t, z_t) - \beta_{t-1}\nabla\ell(x_{t-1}, z_t) \\ &= (\beta_t - \beta_{t-1})\nabla\ell(x_{t-1}, z_t) + \beta_t(\nabla\ell(x_t, z_t) - \nabla\ell(x_{t-1}, z_t)).\end{aligned}$$

Since $\ell$ is $G$-Lipschitz and $H$-smooth, $\|\nabla\ell(x_{t-1}, z_t)\|_* \leq G$ and

$$\|\nabla\ell(x_t, z_t) - \nabla\ell(x_{t-1}, z_t)\|_* \leq H\|x_t - x_{t-1}\| \leq (\beta_t/\beta_{1:t})H\|w_t - x_{t-1}\|,$$

where the second inequality follows from the definition of $x_t$ that $\beta_{1:t}(x_t - x_{t-1}) = \beta_t(w_t - x_{t-1})$. The first result then follows from Proposition 14.

For the second part, by telescopic sum $\beta_t \nabla \mathcal{L}(x_t) = \sum_{i=1}^{t} \beta_i \nabla \mathcal{L}(x_i) - \beta_{i-1} \nabla \mathcal{L}(x_{i-1})$, and recall that $g_t = \sum_{i=1}^{t} \delta_i$. Therefore,

$$\beta_t \nabla \mathcal{L}(x_t) - g_t = \sum_{i=1}^{t} [\beta_i \nabla \mathcal{L}(x_i) - \beta_{i-1} \nabla \mathcal{L}(x_{i-1})] - [\beta_i \nabla \ell(x_i, z_i) - \beta_{i-1} \nabla \ell(x_{i-1}, z_i)].$$

We denote each summand by $X_i$, and we can check $X_i$ satisfies condition (ii) and (iii). First, since we assume $\nabla \mathcal{L}(w) = \mathbb{E}_z[\nabla \ell(w, z)]$ for all $w$, it holds that $\mathbb{E}[X_i | z_{1:i-1}] = 0$. Second, we decompose $X_i$ in the same way as the first part:

$$\mathbb{E}[\|X_i\|_*^2] = \mathbb{E}[\|(\beta_i - \beta_{i-1})[\nabla(\mathcal{L}(x_{i-1}) - \nabla \ell(x_{i-1}, z_i)]$$
$$+ \beta_i([\nabla \mathcal{L}(x_i) - \nabla \ell(x_i, z_i)] - [\nabla \mathcal{L}(x_{i-1}) - \nabla \ell(x_{i-1}, z_i)])\|_*^2]$$

Recall the assumption of $\sigma_G^2$ and $\sigma_H^2$ and that $\|x_i - x_{i-1}\| \leq \beta_i / \beta_{1:i} \|w_i - x_{i-1}\|$.

$$\leq 2(\beta_i - \beta_{i-1})^2 \sigma_G^2 + 2\beta_i^2 \sigma_H^2 \|x_i - x_{i-1}\|^2$$
$$\leq 2(\beta_i - \beta_{i-1})^2 \sigma_G^2 + 2(\beta_i^2 / \beta_{1:i})^2 \sigma_H^2 \|w_i - x_{i-1}\|^2$$
$$\leq 2(k+1)^2 (\sigma_G^2 + D^2 \sigma_H^2) i^{2k-2}.$$

The last inequality follows from the assumption that $\|w_i - x_{i-1}\| \leq D$ and Proposition 14.

The last part of the theorem is a direct result from Lemma 13:

$$\mathbb{E}[\|\nabla \mathcal{L}(x_t) - g_t\|_*^2] \leq \frac{2}{\lambda} \sum_{i=1}^{t} \mathbb{E}[\|X_i\|_*^2] \leq \frac{4(k+1)^2}{\lambda} (\sigma_G^2 + D^2 \sigma_H^2) t^{2k-1}.$$

$\square$

**Lemma 16.** *Suppose* $\mathbb{E}[R_t] = 0$ *and* $\mathbb{E}[\|R_t\|_*^2] \leq \bar{\sigma}_t^2$, *then*

$$\mathbb{E}[\|\gamma_t\|_*^2] \leq 2(\max_{i \leq t} \bar{\sigma}_i^2) \log_2(2t)/\lambda.$$

*Proof.* By construction, $\gamma_t = \sum_{i \in I_t} R_i$, and $R_i$'s are independent and mean-zero. Therefore, Lemma 13 can be applied, which yields

$$\mathbb{E}[\|\gamma_t\|_*^2] \leq \frac{2}{\lambda} \sum_{i \in I_t} \mathbb{E}[\|R_i\|_*^2] \leq \frac{2}{\lambda} \sum_{i \in I_t} \bar{\sigma}_i^2 \leq 2(\max_{i \leq t} \bar{\sigma}^2) \log_2(2t)/\lambda.$$

The last inequality is from the fact that $|I_t| \leq \log_2(2t)$. $\square$

# B Proofs for RDP (Section 2)

In this section, we prove the tree aggregation mechanism for RDP mechanisms implemented in Algorithm 1 correctly composes individual RDP mechanisms. Before that, we will first prove a general composition theorem for RDP.

## B.1 Advanced Composition for RDP

Throughout this section, we use the subscript $1 : T$ to denote a sequence of $T$ elements. We denote $\mathcal{Z}$ the data space and $Z = (z_1, \ldots, z_T)$, $Z' = (z_1', \ldots, z_T')$ neighboring datasets in $\mathcal{Z}^T$ that differs only at the $q$-th element (i.e., $z_t \neq z_t'$ if and only if $t = q$). We consider RDP mechanisms $F_1, \ldots, F_T$ such that $F_1 : \mathcal{Z}^T \to W$ and $F_t : \mathcal{Z}^T \times W^{t-1} \to W$. We assume that for each $t$, there exists some index set $S_t \subseteq [T]$ such that $F_t$ only depends on $S_t$. Formally, we assume:

**Assumption 7.** *Let* $Z, Z' \in \mathcal{Z}$ *be two neighboring datasets which differs only at the $q$-th element. Each $F_t$ associates with $S_t \subseteq [T]$ such that if $q \notin S_t$, then $F_t(Z, x_{1:t-1}) = F_t(Z', x_{1:t-1})$ for all $x_{1:t-1} \in W^{t-1}$.*

For a fixed norm $\| \cdot \|$, we assume $\mathcal{D}$ is a $(V, \alpha)$-RDP distribution on norm $\| \cdot \|$ (see Definition 4), and we define the sensitivity of $F_t$ w.r.t. $\| \cdot \|$ as $\Delta_t(x_{1:t-1})$, a function of inputs $x_{1:t-1}$:

$$\Delta_t(x_{1:t-1}) = \sup_{|Z - Z'| = 1} \| F_t(Z, x_{1:t-1}) - F_t(Z', x_{1:t-1}) \|_*.$$

We also define the output as $\hat{f}_t = F_t(Z, \hat{f}_{1:t-1}) + \sigma_t \zeta_t$, where $\zeta_t \sim \mathcal{D}$ and $\sigma_t^2 \geq \Delta_t(\hat{f}_{1:t-1})^2 / \rho^2$. In particular, $\sigma_t$ only depends on $\hat{f}_{1:t-1}$ and does not depend on $\hat{f}_{t:T}$, i.e., the future. The pseudo-code of this composition is in Algorithm 2. For simplicity, we assume $F$'s are deterministic mechanisms, but we can extend to random mechanisms by treating the random generator as part of the input.

---

**Algorithm 2** Advanced Composition for RDP Mechanisms

---

1: **Input:** Dataset $Z$; functions $F_1, \ldots, F_T$ with sensitivity $\Delta_1, \ldots, \Delta_T$; $(V, \alpha)$-RDP distribution $\mathcal{D}$; privacy constants $\rho_1, \ldots, \rho_T$.
2: Sample random $\zeta_1 \sim \mathcal{D}$ and compute $\sigma_1^2 \geq \Delta_1^2 / \rho_1^2$.
3: Set $f_1 = F_1(Z)$ and $\hat{f}_1 = f_1 + \sigma_1 \zeta_1$
4: **for** $t = 2, \ldots, T$ **do**
5:     Sample random $\zeta_t \sim \mathcal{D}$ and compute $\sigma_t^2 \geq \Delta_t(\hat{f}_{1:t-1})^2 / \rho_t^2$.
6:     Set $f_t = F_t(Z, \hat{f}_{1:t-1})$ and $\hat{f}_t = f_t + \sigma_t \zeta_t$.
7: **Return** $\hat{f}_1, \ldots, \hat{f}_T$.

---

By definition of RDP distribution, each $\hat{f}_t$ is $(\alpha, \alpha \rho^2 / 2)$-RDP. We will also show that the composition $(\hat{f}_1, \ldots, \hat{f}_T)$ is also RDP.

**Theorem 17.** *We define* $\mathrm{IN}(q) = \{t : q \in S_t\}$ *and* $\mathrm{OUT}(q) = \{t : q \notin S_t\}$. *If* $F_1, \ldots, F_T$ *satisfy Assumption 7, then Algorithm 2 is* $(\alpha, S)$-*RDP, where*

$$S = \max_{q \in [T]} \sum_{t \in \mathrm{IN}(q)} \alpha \rho_t^2 / 2.$$

As an immediate corollary, if we set $\rho_t = \rho$ for all $t$ and define $U = \max_q |\mathrm{IN}(q)|$, then Algorithm 2 is $(\alpha, U \alpha \rho^2 / 2)$-RDP.

*Proof.* Let $Z, Z'$ be any neighboring datasets and assume they differ at $q$, and we denote $\hat{f}_t = F_t(Z, \hat{f}_{1:t-1}) + \sigma_t(\hat{f}_{1:t-1}) \zeta_t$ and $\hat{f}'_t = F_t(Z', \hat{f}'_{1:t-1}) + \sigma_t(\hat{f}'_{1:t-1}) \zeta_t$. In this proof, we use the notation $\sigma_t(\hat{f}_{1:t-1})$ to emphasize that $\sigma_t$ satisfying $\sigma_t^2 \geq \Delta_t(\hat{f}_{1:t-1})^2 / \rho_t^2$ is a function of $\hat{f}_{1:t-1}$.

The probability density of the joint distribution of $\hat{f}_{1:T}$, say $P$, and the density of $\hat{f}'_{1:T}$, say $Q$, are:

$$P(x_{1:T}) = \prod_{t=1}^{T} P_t(x_t | x_{1:t-1}), \quad Q(x_{1:T}) = \prod_{t=1}^{T} Q_t(x_t | x_{1:t-1}),$$

where $P_t(\cdot | x_{1:t-1})$ is the density of $(\hat{f}_t | \hat{f}_{1:t-1} = x_{1:t-1}) = F_t(Z, x_{1:t-1}) + \sigma_t(x_{1:t-1}) \zeta_t$ and $Q_t(\cdot | x_{1:t-1})$ is the density of $F_t(Z', x_{1:t-1}) + \sigma_t(x_{1:t-1}) \zeta_t$. Note that $\sigma_t$ is the same for $P_t$ and $Q_t$.

By definition of Rényi divergence,

$$D_\alpha(P \| Q) = \frac{1}{\alpha - 1} \log \int P(x_{1:T})^\alpha Q(x_{1:T})^{1-\alpha} \, dx_{1:T}. \tag{9}$$

Previous multiplication rule implies that:

$$P(x_{1:T})^\alpha Q(x_{1:T})^{1-\alpha} = \prod_{t \in \mathrm{IN}(q) \sqcup \mathrm{OUT}(q)} \left( P_t(x_t | x_{1:t-1})^\alpha Q_t(x_t | x_{1:t-1})^{1-\alpha} \right)$$

For all $t \in \mathrm{OUT}(q)$ (i.e., $q \notin S_t$), Assumption 7 implies that $F_t(Z, x_{1:t-1}) = F_t(Z', x_{1:t-1})$, so $P_t(\cdot | x_{1:t-1}) = Q_t(\cdot | x_{1:t-1})$ and

$$\int P_t(x_t | x_{1:t-1})^\alpha Q_t(x_t | x_{1:t-1})^{1-\alpha} \, dx_t = \int P_t(x_t | x_{1:t-1}) \, dx_t = 1. \tag{10}$$

The second inequality holds because $P_t(\cdot|x_{1:t-1})$ is a probability density.

On the other hand, for all $t \in \text{IN}(q)$,

$$\sigma_t(x_{1:t-1})^2 \geq \Delta_t(x_{1:t-1})^2/\rho_t^2$$
$$\geq \|F_t(Z, x_{1:t-1}) - F_t(Z', x_{1:t-1})\|_*^2/\rho_t^2,$$

so by Definition 4,

$$D_\alpha(P_t\|Q_t) = \frac{1}{\alpha - 1}\log\int P_t(x|x_{1:t-1})^\alpha Q_t(x|x_{1:t-1})^{1-\alpha}\,dx \leq \alpha\rho_t^2/2.$$

Equivalently,

$$\int P_t(x|x_{1:t-1})^\alpha Q_t(x|x_{1:t-1})^{1-\alpha}\,dx \leq \exp((\alpha-1)\alpha\rho_t^2/2). \tag{11}$$

Note that $P_t, Q_t$ only depend on $x_{1:t-1}$ and *not* on $x_{t:T}$, so we can rearrange the integral in (9) as:

$$\int P(x_{1:T})^\alpha Q(x_{1:T})^{1-\alpha}\,dx_{1:T}$$
$$= \int P_T(x_T|x_{1:T-1})^\alpha Q_T(x_T|x_{1:T-1})^{1-\alpha}\cdots\left(\int P_1(x_1)^\alpha Q_1(x_1)^{1-\alpha}\,dx_1\right)\cdots dx_T$$

Evaluating the composite integral from inside to outside with (10) and (11) gives:

$$\leq \exp\left(\sum_{t\in\text{IN}(q)}(\alpha-1)\alpha\rho_t^2/2\right).$$

In conclusion, for all $|Z - Z'| = 1$,

$$D_\alpha(P\|Q) = \frac{1}{\alpha-1}\log\int P(x_{1:T})^\alpha Q(x_{1:T})^{1-\alpha}\,dx_{1:T} \leq \max_q\sum_{t\in\text{IN}(q)}\alpha\rho_t^2/2 = S.$$

$\square$

## B.2 Algorithm 1 is RDP

Now we are ready to prove the tree aggregation in Algorithm 1.

**Theorem 3.** *Suppose $\|\cdot\|^2$ is $\lambda$-strongly convex, $W$ is bounded by $D$, $\ell$ is $G$-Lipschitz and $H$-smooth, and $\mathcal{D}$ is a $(V, \alpha)$-RDP distribution. If $\beta_t = t^k$ and $\sigma_t^2$ is as defined in (6), then Algorithm 1 is $(\alpha, \alpha\rho^2/2)$-DP for all datasets $Z$.*

*Proof.* Recall the definition of $I_t$ in Algorithm 1: we define $s_0 = 0$, and $s_i = \max_k\{s_{i-1} + 2^k : 2^k|t - s_{i-1}\}$ until $s_n = t$ for some $n$, and we define $I_t = \{s_1, \ldots, s_n\}$. For example, $I_4 = \{4\}$ and $I_7 = \{4, 6, 7\}$. We then define $S_t = \{s_{n-1} + 1, s_{n-1} + 2, \ldots, t\}$ (e.g., $S_4 = \{1, 2, 3, 4\}$ and $S_7 = \{7\}$). Observe that $\{S_i : i \in I_t\}$ is a partition of $[t]$.

Let $Z, Z'$ be neighboring datasets that differ at the $q$-th element. Define $F_t : \mathcal{Z}^T \times W^{t-1} \to W$ as:

$$F_t(Z, \hat{f}_{1:t-1}) = \sum_{i\in S_t}\delta_i(Z, \hat{f}_{1:t-1})$$
$$= \sum_{i\in S_t}\beta_i\nabla\ell(x_i, z_i) - \beta_{i-1}\nabla\ell(x_{i-1}, z_i),$$

where $x_i$'s are the parameters as defined in Algorithm 1, and $z_i$ is the $i$-th data in $Z$. We then define $\hat{f}_t = F_t(Z, \hat{f}_{1:t-1}) + \sigma_t(\hat{f}_{1:t-1})\tilde{R}_t$ and $\hat{f}'_t = F_t(Z', \hat{f}'_{1:t-1}) + \sigma_t(\hat{f}'_{1:t-1})\tilde{R}_t$, where $\tilde{R}_t \sim \mathcal{D}$.

For simplicity, we denote $\delta_t = \delta_t(Z, \hat{f}_{1:t-1})$ and $\sigma_t = \sigma_t(\hat{f}_{1:t-1})$ (and $\delta'_t, \sigma'_t$ respectively). We also denote $x_i, w_i$ and $x'_i, w'_i$ as parameters w.r.t. $Z, Z'$ respectively. Since $\{S_i : i \in I_t\}$ partitions $[t]$,

$$g_t + \gamma_t = \sum_{i=1}^t\delta_i + \sum_{i\in I_t}R_i = \sum_{i\in I_t}\left(\sum_{j\in S_i}\delta_j + \sigma_i\tilde{R}_i\right) = \sum_{i\in I_t}\hat{f}_i.$$

By construction of Algorithm 1, $x_1, \ldots, x_t$ are determined by $\{g_i + \gamma_i\}_{i=1}^{t-1}$ and equivalently by $\hat{f}_{1:t-1}$. In particular, they do not depend on $f_{t:T}$. Consequently, this implies that (i) if $\hat{f}_{1:t-1} = \hat{f}'_{1:t-1}$ then $x_{1:t} = x'_{1:t}$ and $w_{1:t} = w'_{1:t}$ and (ii) if in addition $z_i = z'_i$ then $\delta_i = \delta'_i$.

This implies that $F_t$'s satisfy Assumption 7: if $q \notin S_t$ (i.e., $z_i = z'_i$ for all $i \in S_t$), then for any fixed $f_{1:t-1} \in W^{t-1}$, $F_t(Z, f_{1:t-1}) = F_t(Z', f_{1:t-1})$ because $\delta_i = \delta'_i$ for all $i \in S_t$. Consequently, Theorem 17 can be applied, which states that if $\sigma_t(\hat{f}_{1:t-1})^2 \geq \Delta_t(\hat{f}_{1:t-1})^2/\rho^2$, then $(\hat{f}_1, \ldots, \hat{f}_T)$ is $(\alpha, U\alpha\rho^2/2)$-RDP where $U = \max_q |\text{IN}(q)|$ and $\text{IN}(q) = \{t : q \in S_t\}$. Note that $U \leq \log_2(2T)$.

The sensitivity of $F_t$ at fixed $f_{1:t-1} \in W^{t-1}$ is bounded by:

$$\Delta_t(f_{1:t-1}) = \sup_{|Z-Z'|=1} \|F_t(Z, f_{1:t-1}) - F_t(Z', f_{1:t-1})\|_*$$
$$= \sup_{q \in S_t} \|\delta_q - \delta'_q\|_* \leq \sup_{q \in S_t} \|\delta_q\|_* + \sup_{q \in S_t} \|\delta'_q\|_*$$

We proved that $\|\delta_i\|_* \leq (k+1)i^{k-1}(G+H\|w_i-x_{i-1}\|)$ and $\|\delta'_i\|_* \leq (k+1)i^{k-1}(G+H\|w'_i-x'_{i-1}\|)$ (Lemma 15). Note that $w_i = w'_i$ and $x_i = x'_i$ for all $i \leq t$. Also note that $i \leq t$ for all $i \in S_t$, so:

$$\leq 2(k+1)t^{k-1}(G + H\max_{i \in [t]}\|w_i - x_{i-1}\|).$$

Since $U \leq \log_2(2T)$ and $\sigma_t^2$ as defined in (6) satisfies the condition $\sigma_t^2 \geq \Delta_t(\hat{f}_{1:t-1})\log_2(2T)/\rho^2$, Theorem 17 and post-processing imply that $\{g_t + \gamma_t\}_{t=1}^T$ is $(\alpha, \alpha\rho^2/2)$-RDP (so is Algorithm 1). $\quad\square$

## C  Further Discussions about Differential Privacy

### C.1  Example of RDP Distribution

In this subsection, we prove that the multivariate Gaussian distribution $\mathcal{N}(0, I)$ is a $(d, \alpha)$-RDP distribution w.r.t. 2-norm on $\mathbb{R}^d$ for all $\alpha > 0$ (Definition 4). Namely, $\mathcal{N}(0, I)$ satisfies the following three properties: let $R \sim \mathcal{N}(0, I)$, then (i) $\mathbb{E}[R] = 0$, (ii) $\mathbb{E}[\|R\|_2^2] \leq d$, and (iii) for all $\rho > 0$ and $\mu, \mu' \in \mathbb{R}^d$, if $\sigma^2 \geq \|\mu - \mu'\|_2^2/\rho^2$ then $D_\alpha(P\|Q) \leq \alpha\rho^2/2$, where $P, Q$ denote the distribution of $\sigma R + \mu$ and $\sigma R + \mu'$ respectively.

The first property follows immediately from the definition of $\mathcal{N}(0, I)$. For the second property, $R = (r_1, \ldots, r_d)$ where $r_i \sim N(0, 1)$ iid., so $\mathbb{E}[\|R\|_2^2] = \sum_{i=1}^d \mathbb{E}[r_i^2] = d$. To check the third property, we need the following lemma:

**Lemma 18.** $D_\alpha(\mathcal{N}(\mu, \sigma^2 I)\|\mathcal{N}(\mu', \sigma^2 I)) = \alpha\|\mu - \mu'\|^2/2\sigma^2.$

Consequently, for all $\sigma^2 \geq \|\mu - \mu'\|_2^2/\rho^2$, $D_\alpha(\mathcal{N}(0, \sigma^2 I)\|\mathcal{N}(\mu, \sigma^2 I)) \leq \alpha\rho^2/2$. This proves that $\mathcal{N}(0, I)$ is indeed a $(d, \alpha)$-RDP distribution.

*Proof of Lemma 18.* The density of $\mathcal{N}(\mu, \sigma^2 I)$ is $(2\pi\sigma^2)^{-d/2}\exp(-\|x - \mu\|_2^2/2\sigma^2)$. For short we denote $A = (2\pi\sigma^2)^{-d/2}$ and $B = 1/2\sigma^2$. Then

$$D_\alpha(\mathcal{N}(\mu, \sigma^2 I)\|\mathcal{N}(\mu', \sigma^2 I))$$
$$= \frac{1}{\alpha - 1}\log\left(\int_{\mathbb{R}^d} A^\alpha\exp(-B\alpha\|x - \mu\|_2^2)A^{1-\alpha}\exp(-B(1-\alpha)\|x - \mu'\|_2^2)\,dx\right)$$
$$= \frac{1}{\alpha - 1}\log\left(\int_{\mathbb{R}^d} A\exp\left(-B(\alpha\|x - \mu\|_2^2 + (1-\alpha)\|x - \mu'\|_2^2)\right)dx\right).$$

Next, observe that

$$x - \mu = (x - \alpha\mu - (1-\alpha)\mu') - (1-\alpha)(\mu - \mu'),$$
$$x - \mu' = (x - \alpha\mu - (1-\alpha)\mu') + \alpha(\mu - \mu').$$

Consequently, upon expanding out $\|x - \mu\|_2^2, \|x - \mu'\|_2^2$, we get:

$$\alpha\|x - \mu\|_2^2 + (1-\alpha)\|x - \mu'\|_2^2 = \|x - \alpha\mu - (1-\alpha)\mu'\|_2^2 + \alpha(1-\alpha)\|\mu - \mu'\|_2^2.$$

Note that $A \exp(-B\|x - \alpha\mu - (1-\alpha)\mu'\|_2^2)$ is the density of $\mathcal{N}(\alpha\mu + (1-\alpha)\mu', \sigma^2 I)$, so it integrates to 1. Therefore,

$$D_\alpha(\mathcal{N}(\mu, \sigma^2 I) \| \mathcal{N}(\mu', \sigma^2 I))$$

$$= \frac{1}{\alpha - 1} \log \left( \int_{\mathbb{R}^d} A \exp(-B\|x - \alpha\mu - (1-\alpha)\mu'\|_2^2) \exp(-B\alpha(1-\alpha)\|\mu - \mu'\|_2^2) \, dx \right)$$

$$= \frac{1}{\alpha - 1} \log \left( \exp(-B\alpha(1-\alpha)\|\mu - \mu'\|_2^2) \right) = B\alpha \|\mu - \mu'\|_2^2.$$

Recall that $B = 1/2\sigma^2$, and this completes the proof. $\qquad\square$

### C.2 Extension to Pure-DP Mechanisms

In the main text, we focus on Renyi differential privacy, and we defined RDP-distribution (Definition 4) accordingly. We can always extend our result in a pure differential privacy setting.

**Definition 5** (V-DP distribution). *A distribution $\mathcal{D}$ on $\mathbb{R}^d$ is said to be a DP distribution on norm $\|\cdot\|$ with variance constant $V$ (or simply $\mathcal{D}$ is $V$-DP on $\|\cdot\|$) if it satisfies that for $R \sim \mathcal{D}$ (i) $\mathbb{E}[R] = 0$, (ii) $\mathbb{E}[\|R\|_*^2] \leq V$, and (iii) for all $\epsilon > 0$ and $\mu, \mu' \in \mathbb{R}^d$, if $\sigma^2 \geq \|\mu - \mu'\|_*^2/\epsilon^2$, then $p((x - \mu)/\sigma)/p((x - \mu')/\sigma) \leq \exp(\epsilon)$ for all $x \in \mathbb{R}^d$, where $p(x)$ is the density of $\mathcal{D}$.*

The tree aggregation described in Appendix B also works for pure DP mechanisms as well. Therefore, if we assume $\mathcal{D}$ in Algorithm 1 with a $V$-DP distribution and change the definition of $\sigma_t^2$ in (6) correspondingly, Algorithm 1 can be modified to an purely $\epsilon$-DP mechanism.

Next, we can show that exponential mechanism in general norm satisfies this definition:

**Theorem 19.** *Consider a probability density $p(x) = A \exp(-\|x\|_*)$ on $(\mathbb{R}^d, \|\cdot\|)$, where $A$ is some normalization constant. Also define $V = \int_{\mathbb{R}^d} \|x\|_*^2 A \exp(-\|x\|_*) \, dx$. Then distribution $\mathcal{D}$ with density $p$ is a $\mathcal{K}$-DP distribution.*

*Proof.* Let $R \sim \mathcal{D}, \mu, \mu' \in \mathbb{R}^d$, and $\sigma^2 \geq 0$. Since the density $p$ is symmetric, $\mathbb{E}[R] = 0$; and by definition, $\mathbb{E}[\|R\|_*^2] = V$. For the third property,

$$\frac{p((x - \mu)/\sigma)}{p((x - \mu')/\sigma)} = \frac{A \exp(-\|x - \mu\|_*/\sigma)}{A \exp(-\|x - \mu'\|_*/\sigma)} = \exp \left( \frac{-\|x - \mu\|_* + \|x - \mu'\|_*}{\sigma} \right)$$

By triangular inequality, $-\|x - \mu\|_* + \|x - \mu'\|_* \leq \|\mu - \mu'\|_*$, so:

$$\leq \exp \left( \frac{\|\mu - \mu'\|_*}{\sigma} \right).$$

Hence, for all $\sigma \geq \|\mu - \mu'\|_*/\epsilon$, this is further bounded by $\exp(\epsilon)$. $\qquad\square$

## D   Proofs for the Optimistic Case (Section 3)

**Theorem 6.** *Suppose Assumption 1 - 4 hold, and $\mathcal{D}$ is a $(V, \alpha)$-RDP distribution. Set $\beta_t = t^k$ and $\sigma_t^2$ as defined in (6), If the online learner is optimistic (satisfying (8)) with $t$-th gradient $\bar{g}_t = g_t + \gamma_t$ and $t$-th hint $\hat{g}_t = \bar{g}_{t-1}$, then*

$$\frac{\mathbb{E}[\text{Regret}_T(x^*)]}{\beta_{1:T}} \leq O \left( D(G + DH) \left( 1 + \frac{\sqrt{V} \log T}{\sqrt{\lambda}\rho} \right) \frac{1}{T^{3/2}} \right).$$

*Proof.* Recall that $g_t = \sum_{i=1}^t \delta_i$, then

$$\|\bar{g}_t - \hat{g}_t\|_*^2 = \|\delta_t + \gamma_t - \gamma_{t-1}\|_*^2 \leq 3\|\delta_t\|_*^2 + 3\|\gamma_t\|_*^2 + 3\|\gamma_{t-1}\|_*^2.$$

We showed (Lemma 15) that

$$\|\delta_t\|_* \leq (k+1)(G + H\|w_t - x_{t-1}\|)t^{k-1} \leq (k+1)(G + DH)t^{k-1}.$$

Also recall the bound of $\mathbb{E}[\|\gamma_t\|_*^2]$ in (7), so:

$$\mathbb{E}[\|\bar{g}_t - \hat{g}_t\|_*^2] \le 3(k+1)^2(G+DH)^2 t^{2k-2} + \frac{48(k+1)^2 V(G+DH)^2}{\lambda\rho^2}\log_2^2(2T)t^{2k-2}$$

$$= 3(k+1)^2(G+DH)^2 t^{2k-2}\left(1 + \frac{16V\log_2^2(2T)}{\lambda\rho^2}\right).$$

Recall that $\mathbb{E}[\text{Regret}_T(x^*)] \le O(\mathbb{E}[D\sqrt{\sum_{t=1}^{T}\|\bar{g}_t - \hat{g}_t\|_*^2}])$. By Jensen's inequality,

$$\mathbb{E}\left[D\sqrt{\sum_{t=1}^{T}\|\bar{g}_t - \hat{g}_t\|_*^2}\right] \le D\sqrt{\sum_{t=1}^{T}\mathbb{E}[\|\bar{g}_t - \hat{g}_t\|_*^2]}$$

$$\le \sqrt{3}(k+1)D(G+DH)\left(1 + \frac{4\sqrt{V}\log_2(2T)}{\sqrt{\lambda}\rho}\right)T^{k-1/2}.$$

Finally, dividing this bound by $\beta_{1:T} \ge T^{k+1}/(k+1)$ completes the proof. $\qquad\square$

## E   Proofs for the Strongly Convex Case (Section 4)

**Lemma 20.** *For any sequence $\beta_t > 0, g_t \in \mathbb{R}^d$, suppose an online learner predicts $w_t$ and receives $t$-th loss $\ell_t(w) = \langle g_t, w\rangle$, and define $x_t = \sum_{i=1}^{t}\frac{\beta_i w_i}{\beta_{1:t}}$. If $\mathcal{L}$ is $\mu$-strongly convex w.r.t. $\|\cdot\|$, then*

$$\beta_{1:T}(\mathcal{L}(x_T) - \mathcal{L}(x^*)) \le \text{Regret}_T(x^*) + \sum_{t=1}^{T}\left(\langle\beta_t\nabla\mathcal{L}(x_t) - g_t, w_t - x^*\rangle - \frac{\beta_t\mu}{2}\|x_t - x^*\|^2\right).$$

*Proof.* We start with the strong convexity identity $\mathcal{L}(x^*) \ge \mathcal{L}(x_t) + \langle\nabla\mathcal{L}(x_t), x^*-x_t\rangle + \frac{\mu}{2}\|x_t-x^*\|^2$:

$$\sum_{t=1}^{T}\beta_t(\mathcal{L}(x_t) - \mathcal{L}(x^*)) \le \sum_{t=1}^{T}\beta_t\langle\nabla\mathcal{L}(x_t), x_t - x^*\rangle - \frac{\beta_t\mu}{2}\|x_t - x^*\|^2. \qquad(12)$$

With the same argument in the proof of Lemma 1, we can show:

$$\beta_t\langle\nabla\mathcal{L}(x_t), x_t - x^*\rangle = \beta_t\langle\nabla\mathcal{L}(x_t), x_t - w_t\rangle + \beta_t\langle\nabla\mathcal{L}(x_t), w_t - x^*\rangle$$

Recall the definition that $\beta_{1:t}x_t = \beta_{1:t-1}x_{t-1} + \beta_t w_t$ and thus $\beta_t(x_t - w_t) = \beta_{1:t-1}(x_{t-1} - x_t)$. Also, since $\mathcal{L}$ is convex, $\langle\nabla\mathcal{L}(x_t), x_{t-1} - x_t\rangle \le \mathcal{L}(x_{t-1}) - \mathcal{L}(x_t)$, so:

$$\le \beta_{1:t-1}\mathcal{L}(x_{t-1}) - \beta_{1:t-1}\mathcal{L}(x_t) + \langle\beta_t\nabla\mathcal{L}(x_t) - g_t + g_t, w_t - x^*\rangle.$$

Consequently, moving $\sum_{t=1}^{T}\beta_t\mathcal{L}(x_t)$ to the right side and taking the telescopic sum in (12) gives:

$$-\beta_{1:T}\mathcal{L}(x^*) \le \sum_{t=1}^{T}\beta_t\langle\nabla\mathcal{L}(x_t), x_t - x^*\rangle - \beta_t\mathcal{L}(x_t) - \frac{\beta_t\mu}{2}\|x_t - x^*\|^2$$

$$\le -\beta_{1:T}\mathcal{L}(x_T) + \text{Regret}_T(x^*) + \sum_{t=1}^{T}\langle\beta_t\nabla\mathcal{L}(x_t) - g_t, w_t - x^*\rangle - \frac{\beta_t\mu}{2}\|x_t - x^*\|^2.$$

Moving $\beta_{1:T}\mathcal{L}(x_T)$ to the left completes the proof. $\qquad\square$

This lemma immediately implies Lemma 7.

**Lemma 7.** *Suppose $\mathcal{L}$ is $\mu$-strongly convex w.r.t. $\|\cdot\|$. If we replace $\ell_t(w) = \langle g_t + \gamma_t, w\rangle$ with $\bar{\ell}_t(w) = \ell_t(w) + \frac{\beta_t\mu}{4}\|w - x_t\|^2$ in Algorithm 1, and denote the associated regret by $\overline{\text{Regret}}_T$, then*

$$\beta_{1:T}(\mathcal{L}(x_T) - \mathcal{L}(x^*)) \le \overline{\text{Regret}}_T(x^*) + \sum_{t=1}^{T}\langle\beta_t\nabla\mathcal{L}(x_t) - g_t - \gamma_t, w_t - x^*\rangle - \frac{\beta_t\mu}{8}\|w_t - x^*\|^2$$

$$\le \overline{\text{Regret}}_T(x^*) + \sum_{t=1}^{T}\frac{2\|\beta_t\nabla\mathcal{L}(x_t) - g_t - \gamma_t\|_*^2}{\beta_t\mu}.$$

*Proof.* By definition, $\bar{\ell}_t(w) = \ell_t(w) + \frac{\beta_t \mu}{4} \|w - x_t\|^2$, so

$$\overline{\text{Regret}}_T(x^*) = \sum_{t=1}^{T} \left( \ell_t(w_t) + \frac{\beta_t \mu}{4} \|w_t - x_t\|^2 \right) - \left( \ell_t(x^*) + \frac{\beta_t \mu}{4} \|x^* - x_t\|^2 \right)$$

$$= \text{Regret}_T(x^*) + \sum_{t=1}^{T} \frac{\beta_t \mu}{4} (\|w_t - x_t\|^2 - \|x_t - x^*\|^2).$$

Upon substituting this equation into Lemma 20, we get:

$$\beta_{1:T}(\mathcal{L}(x_T) - \mathcal{L}(x^*))$$

$$\leq \overline{\text{Regret}}_T(x^*) - \sum_{t=1}^{T} \frac{\beta_t \mu}{4} (\|w_t - x_t\|^2 - \|x_t - x^*\|^2)$$

$$+ \sum_{t=1}^{T} \langle \beta_t \nabla \mathcal{L}(x_t) - g_t - \gamma_t, w_t - x^* \rangle - \frac{\beta_t \mu}{2} \|x_t - x^*\|^2$$

$$\leq \overline{\text{Regret}}_T(x^*) + \sum_{t=1}^{T} \langle \beta_t \nabla \mathcal{L}(x_t) - g_t - \gamma_t, w_t - x_t \rangle - \frac{\beta_t \mu}{4} (\|w_t - x_t\|^2 + \|x_t - x^*\|^2)$$

$$\leq \overline{\text{Regret}}_T(x^*) + \sum_{t=1}^{T} \langle \beta_t \nabla \mathcal{L}(x_t) - g_t - \gamma_t, w_t - x^* \rangle - \frac{\beta_t \mu}{8} \|w_t - x^*\|^2.$$

The last inequality follows from the identity $\|w_t - x^*\|^2 \leq 2\|w_t - x_t\|^2 + 2\|x_t - x^*\|^2$.

For the second inequality in the lemma, by Fenchel-Young's inequality,

$$\langle \beta_t \nabla \mathcal{L}(x_t) - g_t - \gamma_t, w_t - x^* \rangle - \frac{\beta_t \mu}{8} \|w_t - x^*\|^2$$

$$\leq \|\beta_t \nabla \mathcal{L}(x_t) - g_t - \gamma_t\|_* \|w_t - x^*\| - \frac{\beta_t \mu}{8} \|w_t - x^*\|^2$$

For any quadratic of form $ax - bx^2$ and $a, b > 0$, note that $\sup_x ax - bx^2 \leq a^2/4b$, so:

$$\leq \frac{2\|\beta_t \nabla \mathcal{L}(x_t) - g_t - \gamma_t\|_*^2}{\beta_t \mu}.$$

$\square$

**Proposition 21.** *Suppose $W$ is a convex bounded domain with diameter $D$, and let $u \in W$ and $f(w) = \|w - u\|^2$. Then for all $w \in W$ and $v \in \partial f(w)$, $\|v\|_* \leq 2D$.*

*Proof.* Let $\phi(r) = r^2$ and $g(w) = \|w - u\|$, then $f(w) = \phi \circ g(w)$. By chain rule of sub-differentials (Corollary 16.72 Bauschke et al. (2011)),

$$\partial f(w) = \{\alpha v' : \alpha \in \partial \phi(g(w)), v' \in \partial g(w)\}$$

$$= \{2\|w - u\| v' : v' \in \partial \|w - u\|\}.$$

By assumption, $\|w - u\| \leq D$. Moreover, $\|\cdot\|$ is 1-Lipschitz (because $\|x\| - \|y\| \leq \|x - y\|$), so $\|v'\|_* \leq 1$ for all $v' \in \partial \|w - u\|$. As a result, for all $v \in \partial f(w)$, $\|v\|_* = 2\|w - u\| \|v'\|_* \leq 2D$. $\square$

## F   Proofs for the Parameter-free Case (Section 5)

**Definition 6.** A random vector $X \in \mathbb{R}^d$ is said to be $\sigma$-norm-sub-Gaussian, denoted by $\text{nSG}(\sigma)$ if

$$\mathcal{P}\{\|X - \mathbb{E}[X]\|_2 \geq \epsilon\} \leq 2 \exp\left( -\frac{\epsilon^2}{2\sigma^2} \right), \forall \epsilon.$$

We will rely on the following concentration bound on norm-sub-Gaussian random vectors.

**Lemma 22** (Lemma 1, Jin et al. (2019)). *There exists a universal $C$ such that (i) if $\|X\| \leq \sigma$, then $X$ is $\mathrm{nSG}(C\sigma)$ nad (ii) if $X$ is $\sigma$-sub-Gaussian, then $X$ is $\mathrm{nSG}(C\sqrt{d}\sigma)$.*

**Lemma 23** (Corollary 8, Jin et al. (2019)). *There exists a universal constant $C$ such that if $X_i|X_{1:i-1}$ is mean-zero $\mathrm{nSG}(\sigma_i)$ for all $X_1, \ldots, X_t$, then for any fixed $\delta > 0$ and $B > b > 0$ such that $b < \sum_{i=1}^{t} \sigma_i^2 \leq B$ almost surely, with probability at least $1 - \delta$,*

$$\left\| \sum_{i=1}^{t} X_i \right\|_2 \leq C \sqrt{\sum_{i=1}^{t} \sigma_i^2 \left( \log \frac{2d}{\delta} + \log \log \frac{B}{b} \right)}.$$

Recall that $\beta_t \nabla \mathcal{L}(x_t) - g_t = \sum_{i=1}^{t} X_i$ (Lemma 15), where

$$X_i = [\beta_i \nabla \mathcal{L}(x_i) - \beta_{i-1} \nabla \mathcal{L}(x_{i-1})] - [\beta_i \nabla \ell(x_i, z_i) - \beta_{i-1} \nabla \ell(x_{i-1}, z_i)];$$

and $\gamma_t = \sum_{i \in I_t} R_i$, where $R_i = \sigma_i \tilde{R}_i$ and $R_i \sim \mathcal{D}$ i.i.d. We have the following lemma:

**Lemma 24.** *Suppose Assumption 2 - 4 hold w.r.t. the 2-norm, and suppose $\mathcal{D}$ is a $(V, \alpha)$-RDP distribution and is $\sigma_{\mathcal{D}}$-sub-Gaussian, i.e.,*

$$\mathcal{P}\{ \sup_{\|a\|=1} \langle X, a \rangle \geq \epsilon \} \leq \exp\left( -\frac{\epsilon^2}{2\sigma_{\mathcal{D}}^2} \right).$$

*Also set $\beta_t = t^k$. Then there exists a universal constant $C$ such that $X_i|X_{1:i-1}$ are mean-zero $\mathrm{nSG}(\sigma_{X_i})$ and $R_i|R_{1:i-1}$ are mean-zero $\mathrm{nSG}(\sigma_{R_i})$ for all $i$, where*

$$\sigma_{X_i} = 2C(k+1)(G + H\|w_i - x_{i-1}\|_2)i^{k-1},$$
$$\sigma_{R_i} = C\sqrt{d}\sigma_{\mathcal{D}}\sigma_i.$$

*Proof.* Since we assume $\mathbb{E}[\nabla \ell(x, z)] = \nabla \mathcal{L}(x)$ for all $x$, $\mathbb{E}[X_i|X_{1:i-1}] = 0$. Also, since $\mathcal{D}$ is a $(V, \alpha)$-RDP distribution (Definition 4) and $R_i = \sigma_i \tilde{R}_i$'s are independent, $\mathbb{E}[R_i|R_{1:i-1}] = \mathbb{E}[\tilde{R}_i] = 0$.

For the second part, in Lemma 15 we proved that

$$\|\delta_i\|_2 = \|\beta_i \nabla \ell(x_i, z_i) - \beta_{i-1} \nabla \ell(x_{i-1}, z_i)\|_2 \leq (k+1)(G + H\|w_i - x_{i-1}\|_2)i^{k-1}.$$

The same bound holds for $\beta_i \nabla \mathcal{L}(x_i) - \beta_{i-1} \nabla \mathcal{L}(x_{i-1})$ following the same argument. Therefore,

$$\|X_i\|_2 \leq 2(k+1)(G + H\|w_i - x_{i-1}\|_2)i^{k-1}.$$

Moreover, since we assume $\tilde{R}_i \sim \mathcal{D}$ is $\sigma_{\mathcal{D}}$-sub-Gaussian, $R_i = \sigma_i \tilde{R}_i$ is $\sigma_i \sigma_{\mathcal{D}}$-sub-Gaussian. Hence, by Lemma 22, $X_i|X_{1:i-1}$ and $R_i|R_{1:i-1}$ are norm-sub-Gaussian. $\qquad\square$

**Theorem 10.** *Suppose w.r.t. 2-norm, $W$ is bounded by $D$ and $\ell$ is $G$-Lipschitz and $H$-smooth. Suppose $\mathcal{D}$ is $(V, \alpha)$-RDP distribution and is $\sigma_{\mathcal{D}}$-sub-Gaussian. If we set $\beta_t = t^3$ (i.e. $k = 3$) and set $\sigma_t^2$ as defined in (6), then with probability at least $1 - \delta$,*

$$\mathcal{L}(x_T) - \mathcal{L}(x^*) \leq \frac{4}{T^4} \left( \mathrm{Regret}_T(x^*) + \sum_{t=1}^{T} \xi_t(\|w_t\|_2 + \|x^*\|_2) + \nu_t(\|w_t\|_2^2 + \|x^*\|_2^2) \right).$$

*where $C$ is a universal constant, $A = 8\sqrt{2}C^2$, $A' = 8\sqrt{d}\sigma_{\mathcal{D}}C^2$, $\kappa = 1 + DH/G$, and*

$$\xi_t = AG\Phi t^{5/2} + A'(G + DH)\frac{\Phi \log_2(2T)t^2}{\rho}, \; \nu_t = 28AH\Phi t^{5/2}, \; \Phi = \sqrt{\log \frac{20dT\log(2\kappa T)}{\delta}}.$$

*Proof.* We start with Eq. (2):

$$\beta_{1:T}(\mathcal{L}(x_T) - \mathcal{L}(x^*)) \leq R_T(x^*) + \sum_{t=1}^{T} \langle \beta_t \nabla \mathcal{L}(x_t) - g_t - \gamma_t, w_t - x^* \rangle$$

$$\leq R_T(x^*) + \sum_{t=1}^{T} \left( \|\beta_t \nabla \mathcal{L}(x_t) - g_t\|_2 + \|\gamma_t\|_2 \right) \left( \|w_t - x^*\|_2 \right). \tag{13}$$

**Step 1.** By Lemma 23 and 24, for each $t$, with probability $1 - \delta/2T$,

$$\|\beta_t \nabla \mathcal{L}(x_t) - g_t\|_2 \leq C \sqrt{\sum_{i=1}^{t} \sigma_{X_i}^2 \left( \log \frac{4dT}{\delta} + \log \log \frac{B}{b} \right)}.$$

Since we choose $\beta_t = t^3$ (i.e., $k = 3$),

$$\sigma_{X_i} = 8Ci^2(G + H\|w_i - x_{i-1}\|_2).$$

Next, we can bound $\sum_{i=1}^{t} \sigma_{X_i}^2$ as follows: for all $t$,

$$\sum_{i=1}^{t} \sigma_{X_i}^2 \leq B := 64C^2(G + DH)^2 T^5,$$

$$\sum_{i=t}^{t} \sigma_{X_i}^2 \geq b := \sigma_{X_1}^2 = 64C^2(G + H\|w_1\|_2)^2.$$

Recall that $\kappa = 1 + DH/G$, so

$$\frac{B}{b} = \frac{(G + DH)^2 T^5}{(G + H\|w_1\|_2)^2} \leq (\kappa T)^5.$$

Also recall that $\Phi = \sqrt{\log(20dT \log(2\kappa T)/\delta)}$, so

$$\sqrt{\log \frac{4dT}{\delta} + \log \log \frac{B}{b}} \leq \sqrt{\log \frac{20dT \log(\kappa T)}{\delta}} \leq \Phi.$$

Therefore, with probability at least $1 - \delta/2T$,

$$\|\beta_t \nabla \mathcal{L}(x_t) - g_t\|_2 \leq C\Phi \sqrt{\sum_{i=1}^{t} [8Ci^2(G + H\|w_i - x_{i-1}\|_2)]^2}$$

$$\leq 8C^2\Phi \sqrt{\sum_{i=1}^{t} i^4(G + H\|w_i - x_{i-1}\|_2)^2}. \tag{14}$$

By union bound, with probability at least $1 - \delta/2$,

$$\sum_{t=1}^{T} \|\beta_t \nabla \mathcal{L}(x_t) - g_t\|_2 \|w_t - x^*\|_2$$

$$\leq 8C^2\Phi \sum_{t=1}^{T} \sqrt{\sum_{i=1}^{t} i^4(G + H\|w_i - x_{i-1}\|_2)^2} \|w_t - x^*\|_2$$

We use the identity $(a + b)^2 \leq 2a^2 + 2b^2$ and $\sqrt{a + b} \leq \sqrt{a} + \sqrt{b}$:

$$\leq 8C^2\Phi \sum_{t=1}^{T} \left( \sqrt{\sum_{i=1}^{t} 2G^2 i^4} + \sqrt{\sum_{i=1}^{t} 2H^2\|w_i - x_{i-1}\|_2^2 i^4} \right) \|w_t - x^*\|_2 \tag{15}$$

**1.1.** We bound these two sums separately. For the first sum, recall that $A = 8\sqrt{2}C^2$, so

$$8C^2 \sum_{t=1}^{T} \sqrt{\sum_{i=1}^{t} 2G^2 i^4} \|w_t - x^*\|_2 \leq AG \sum_{t=1}^{T} t^{5/2}(\|w_t\|_2 + \|x^*\|_2).$$

**1.2.** For the second sum, we apply Young's inequality ($ab \leq \frac{1}{2\lambda}a^2 + \frac{\lambda}{2}b^2$) for each $t$:

$$8C^2 \sum_{t=1}^{T} \sqrt{\sum_{i=1}^{t} 2H^2\|w_i - x_{i-1}\|_2^2 i^4} \|w_t - x^*\|_2$$

$$\leq AH \sum_{t=1}^{T} \frac{1}{2\lambda_t} \sum_{i=1}^{t} (\|w_i - x_{i-1}\|_2^2 i^4) + \frac{\lambda_t}{2}\|w_t - x^*\|_2^2$$

We first bound $\|w_i - x_{i-1}\|_2^2 \leq 2\|w_i\|_2^2 + 2\|x_{i-1}\|_2^2$. Recall that $x_0 = 0$ and for $i \geq 2$, $x_{i-1} = \sum_{j=1}^{i-1} \frac{\beta_j}{\beta_{1:i-1}} w_j$, so $\|x_{i-1}\|_2^2 \leq \sum_{j=1}^{i-1} \frac{\beta_j}{\beta_{1:i-1}} \|w_j\|_2^2$ by convexity. Consequently,

$$\leq AH \sum_{t=1}^{T} \left( \frac{1}{\lambda_t} \sum_{i=1}^{t} i^4 \|w_i\|_2^2 + \frac{1}{\lambda_t} \sum_{i=2}^{t} i^4 \sum_{j=1}^{i-1} \frac{\beta_j \|w_j\|_2^2}{\beta_{1:i-1}} + \lambda_t(\|w_t\|_2^2 + \|x^*\|_2^2) \right). \quad (16)$$

We define $\lambda_t = ct^{5/2}$ for some constant $c$ to be determined later, and we apply change of summation on the first two sums:

**Lemma 25.** *For any sequence $a_i, b_j, c_k$,*

$$\sum_{i=1}^{N} a_i \sum_{j=1}^{i} b_j = \sum_{i=1}^{N} b_i \sum_{j=i}^{N} a_j, \quad and \quad \sum_{i=1}^{N} a_i \sum_{j=1}^{i} b_j \sum_{k=1}^{j} c_k = \sum_{i=1}^{N} c_i \sum_{j=i}^{N} a_j \sum_{k=i}^{j} b_k.$$

**1.2.1.** For the first summation,

$$\sum_{t=1}^{T} \frac{1}{\lambda_t} \sum_{i=1}^{t} i^4 \|w_i\|_2^2 = \sum_{t=1}^{T} \sum_{i=t}^{T} \frac{1}{\lambda_i} t^4 \|w_t\|_2^2$$

For decreasing function $f$, $\sum_{i=t+1}^{T} f(i) \leq \int_t^T f(x)\, dx$, then:

$$\leq \sum_{t=1}^{T} \left( \frac{1}{ct^{5/2}} + \int_t^{\infty} \frac{1}{cx^{5/2}}\, dx \right) t^4 \|w_t\|_2^2 \leq \sum_{t=1}^{T} \frac{5}{3c} t^{5/2} \|w_t\|_2^2.$$

**1.2.2.** For the second term, by Proposition 14, $\beta_{1:i-1} \geq (i-1)^4/4$, so

$$\sum_{t=1}^{T} \frac{1}{\lambda_t} \sum_{i=2}^{t} i^4 \sum_{j=1}^{i-1} \frac{\beta_j \|w_j\|_2^2}{\beta_{1:i-1}} \leq \sum_{t=1}^{T} \frac{1}{ct^{5/2}} \sum_{i=2}^{t} \frac{4i^4}{(i-1)^4} \sum_{j=1}^{i} j^3 \|w_j\|_2^2$$

For all $i \geq 2$, we can bound $i/(i-1) \leq 2$. We then apply change of summation, which gives:

$$\leq \sum_{t=1}^{T} \sum_{i=t}^{T} \frac{1}{ci^{5/2}} \sum_{j=t}^{i} 64 t^3 \|w_t\|_2^2 \leq \sum_{t=1}^{T} \frac{192}{c} t^{5/2} \|w_t\|_2^2.$$

The last inequality is again derived from the integral bound:

$$\sum_{i=t}^{T} \frac{1}{i^{5/2}} \sum_{j=t}^{i} 1 \leq \sum_{i=t}^{T} \frac{1}{i^{3/2}} \leq \frac{1}{t^{3/2}} + \int_t^{\infty} \frac{1}{x^{3/2}}\, dx \leq \frac{3}{t^{1/2}}.$$

In conclusion, upon substituting **1.2.1.** and **1.2.2.** into (16) and setting $c = 14$, we get:

$$8C^2 \sum_{t=1}^{T} \sqrt{\sum_{i=1}^{t} 2H^2 \|w_i - x_{i-1}\|_2^2 i^4} \|w_t - x^*\|_2$$

$$\leq AH \sum_{t=1}^{T} \left( \frac{5}{3c} t^{5/2} \|w_t\|_2^2 + \frac{192}{c} t^{5/2} \|w_t\|_2^2 + ct^{5/2}(\|w_t\|_2^2 + \|x^*\|_2^2) \right)$$

$$\leq AH \sum_{t=1}^{T} 28 t^{5/2} (\|w_t\|_2^2 + \|x^*\|_2^2).$$

Moreover, upon substituting **1.1.** and **1.2.** into (15), we get: with probability at least $1 - \delta/2$,

$$\sum_{t=1}^{T} \|\beta_t \nabla \mathcal{L}(x_t) - g_t\|_2 \|w_t - x^*\|_2$$

$$\leq A\Phi \sum_{t=1}^{T} Gt^{5/2}(\|w_t\|_2 + \|x^*\|_2) + 28H t^{5/2}(\|w_t\|_2^2 + \|x^*\|_2^2).$$

**Step 2:** We can bound $\sum_{t=1}^{T}\|\gamma_t\|_2\|w_t - x^*\|_2$ in a similar way. By Lemma 24 and definition of $\sigma_t$ in (6), $R_i|R_{1:i-1}$ is mean-zero $\mathrm{nSG}(\sigma_{R_i})$, where

$$\sigma_{R_i} = C\sqrt{d}\sigma_{\mathcal{D}}\sigma_i = \frac{8\sqrt{d}\sigma_{\mathcal{D}}C}{\rho}\sqrt{\log_2(2T)}(G + H\max_{j\in[i]}\|w_j - x_{j-1}\|_2)i^2.$$

Next, we can bound $\sum_{i\in I_t}\sigma_{R_i}^2$ as follows: for all $t$,

$$\sum_{i\in I_t}\sigma_{R_i}^2 \geq \min_{i\in I_t}\sigma_{R_i}^2 \geq b := \frac{64d\sigma_{\mathcal{D}}^2C^2}{\rho^2}\log_2(2T)G^2.$$

On the other hand, since $|I_t| \leq \log_2(2T)$,

$$\sum_{i\in I_t}\sigma_{R_i}^2 \leq B_t := \frac{64d\sigma_{\mathcal{D}}^2C^2}{\rho^2}\log_2^2(2T)(G + DH)^2t^4.$$

Hence, $B_t/b \leq \log_2(2T)\kappa^2T^4 \leq (2\kappa T)^5$ (because $\log_2(2T) \leq 2T$ and $\kappa \geq 1$). By definition of $\Phi$,

$$\sqrt{\log\frac{4dT}{\delta} + \log\log\frac{B_t}{b}} \leq \sqrt{\log\frac{20dT\log(2\kappa T)}{\delta}} = \Phi.$$

Recall that $A' = 8\sqrt{d}\sigma_{\mathcal{D}}C^2$. By Lemma 23, for each $t$, with probability at least $1 - \delta/2T$,

$$\|\gamma_t\|_2 \leq C\sqrt{\sum_{i\in I_t}\sigma_{R_i}^2\left(\log\frac{4dT}{\delta} + \log\log\frac{B_t}{b}\right)} \leq \frac{A'}{\rho}(G + DH)\Phi\log_2(2T)t^2. \qquad (17)$$

By union bound, with probability at least $1 - \delta/2$,

$$\sum_{t=1}^{T}\|\gamma_t\|_2\|w_t - x^*\|_2 \leq \sum_{t=1}^{T}\frac{A'}{\rho}(G + DH)\Phi\log_2(2T)t^2(\|w_t\|_2 + \|x^*\|_2).$$

In conclusion, we take the union bound on the results from **step 1.** and **step 2.** and substitute it back to the starting point (13). Then with probability at least $1 - \delta$,

$$\beta_{1:T}(\mathcal{L}(x_T) - \mathcal{L}(x^*)) \leq R_T(x^*) + \sum_{t=1}^{T}28AH\Phi t^{5/2}(\|w_t\|_2^2 + \|x^*\|_2^2)$$

$$+ \sum_{t=1}^{T}\left(AG\Phi t^{5/2} + A'(G + DH)\frac{\Phi\log_2(2T)t^2}{\rho}\right)(\|w_t\|_2 + \|x^*\|_2).$$

Define $\xi_t, \nu_t$ as in the theorem, and recall that $\beta_{1:T} \geq T^4/4$. This completes the proof. $\qquad\square$

**Lemma 25.** *For any sequence* $a_i, b_j, c_k$,

$$\sum_{i=1}^{N}a_i\sum_{j=1}^{i}b_j = \sum_{i=1}^{N}b_i\sum_{j=i}^{N}a_j, \quad and \quad \sum_{i=1}^{N}a_i\sum_{j=1}^{i}b_j\sum_{k=1}^{j}c_k = \sum_{i=1}^{N}c_i\sum_{j=i}^{N}a_j\sum_{k=i}^{j}b_k.$$

*Proof.* The proof is basically re-pairing the summations:

$$\sum_{i=1}^{N}\sum_{j=1}^{i}a_ib_j = a_1b_1 + a_2(b_1 + b_2) + a_3(b_1 + b_2 + b_3) + \cdots$$

$$= (a_1 + \cdots + a_N)b_1 + (a_2 + \cdots + a_N)b_2 + \cdots \sum_{i=1}^{T}\sum_{j=0}^{T-i}a_{t-j}b_i.$$

For the second part of the theorem, denote $B_j^i = \sum_{k=j}^i b_k$. By first part,

$$
\begin{aligned}
LHS &= \sum_{i=1}^N a_i \sum_{j=1}^i c_j \sum_{k=j}^i b_k \\
&= \sum_{i=1}^N \sum_{j=1}^i a_i c_j (B_j^N - B_{i+1}^N) \\
&= \sum_{i=1}^N \sum_{j=i}^N a_j c_i B_i^N - a_j c_i B_{j+1}^N \\
&= \sum_{i=1}^N \sum_{j=i}^N a_j c_i B_i^j.
\end{aligned}
$$

We then recover the lemma once we substitute $B_i^j = \sum_{k=i}^j b_k$.  $\square$