# OpenReview forum: "Differentially Private Online-to-batch for Smooth Losses"
_NeurIPS.cc/2022/Conference — NeurIPS 2022 Accept_

### Official Review · Reviewer_ycjh · 2022-07-05

**Rating:** 8
**Confidence:** 4
**Soundness:** 4 excellent
**Presentation:** 3 good
**Contribution:** 4 excellent

**Summary:**

Online to batch conversions are a classical reduction from adversarial online learning to stochastic optimization. This work introduces the use of such reductions in the context of Differentially Private Stochastic Convex Optimization (DP-SCO). This extension is nontrivial, as it is known that in this context classical online to batch leads to suboptimal rates. Therefore, in this work a more careful weighted regret scheme (known as anytime online to batch conversions, Cutkosky 2019) is used. This method is used in conjunction with a tree aggregation DP mechanism (Dwork et al. 2010), that allows for optimal noise addition of partial sums; and with a recursive stochastic gradient estimator, which has been proved useful in DP-SCO (Asi et al. 2021, Bassily et al. 2021). It is interesting that the technique is quite general, and it is applicable to settings beyond $\ell_2$ DP-SCO, which by now is well understood.  The work includes extensions to strongly convex losses and rates for which (the non-private $O(1/\sqrt{n})$ term) adapts to the level of noise.

* Asi, Feldman, Koren, Talwar. Private stochastic convex optimization: Optimal rates in l1 geometry (ICML'21) https://arxiv.org/ abs/2103.01516.
* Bassily, Guzman, Nandi. Non-Euclidean Differentially Private Stochastic Optimization (COLT'21) https://arxiv.org/pdf/2103.01278.pdf
* Cutkosky. Anytime online-to-batch, optimism and acceleration (ICML'19) https://arxiv.org/pdf/1903.00974.pdf
* Dwork, Naor Pitassi, Rothblum. Differential Privacy under Continual Observation (STOC'10)


**Questions:**

1. Assumption 1, page 2. It seems that the strong convexity inequality is incorrectly written. Shouldn't it be $\|y\|^2\geq \|x\|^2+\langle g,y-x\rangle+\frac{\lambda}{2}\|y-x\|^2$? It should also be said that this function is strongly convex with respect to $\|\cdot\|$.
2. Algorithm 1, page 4. The subroutine NOISE is somewhat confusing, and I expect that any reader unfamiliar with the tree aggregation mechanism will have difficulties following it. What is the binary expression of $t$? do you mean ``binary expansion'', and then does this mean the bit encoding? I think a brief description of tree aggregation with references can go a long way. As well as adding a short lemma that elaborates on the connections between the binary encoding, $s_t$, $I_t$, $S_t$, etc. (e.g., that $S_t$'s partition $[t]$).
3. Equation (3). It is unclear to me why you use triangle inequality and Holder here. I thought the point is that these are zero mean conditional random variables (in fact, in the previous page this is exactly what is done in the last paragraph). I am not saying that dependencies on the norms of these vectors would not affect the final rates (because they do), but upper bounding in this way here is misleading to what is ultimately done in the analysis.
4. Main Result, page 5 (and other places). I understand it is possible to choose different values of $k$ for the averaging, but it is never thoroughly discussed why would one want this level of generality. A couple of results use particular choices of $k$, but it is never discussed why this choice matters.
5. Theorem 6, page 6. What is the value of $k$ for this result? What is the importance of the $3/2$ exponent of $T$ in the average weighted regret?
6. Definition of sub-Gaussianity, Page 8. What is $R_i$ here? Reading this feels out of context.
7. Theorem 10, page 8. What is the importance of $k=3$ here?
8. Conclusion, Page 9. It is said that ``we cannot tolerate large $H$.'' This issue is never discussed in the paper, neither its nature or whether it seems inherent. I am afraid Theorem statements neither make this limitation clear. Please elaborate where it corresponds.
9. Line 391-392, Page 13. Please remove $\nabla$ in various terms where they shouldn't be (I believe it was meant to be ${
cal L}$, rather than $\nabla {\cal L}$).
10. Lemma 15, page 14. Say that $\|\cdot\|^2$ must be strongly convex ``with respect to $\|\cdot\|$''. And add the expectation sign to $\|\delta_t\|_{\ast}$ (o.w. it is not true). Also add the conditional expectation to $\mathbb{E}[\|X_i\|_{\ast}^2]$ in (iii), to be consistent with (ii).
11. Proof of Theorem 3. The quick preliminaries of the tree aggregation procedure should be converted into a short Lemma. That would make it easier to find for the interested reader (these notations and claims are used beyond this theorem).
12. Theorem 19, page 19. Is this result ever used?
13. Line 546, page 20. Use the expectation operator on $\|\delta_t\|_{\ast}$.
14. Could the authors elaborate on the comparison with a recent work on DP-SCO with streaming data: Han et al. 2022 (On Private Online Convex Optimization: Optimal Algorithms in lp-Geometry and High Dimensional Contextual Bandits, https://arxiv.org/pdf/2206.08111.pdf). Maybe the results of the current submission subsume some of these results?


**Limitations:**

There are two technical aspects of this work that are worth a more extensive discussion:

1. Are the new online to batch conversion results presented in this paper capable of addressing $\ell_1$-geometry (Asi et al. 2021, Bassily et al. 2021). This is an important question, since this is the only general case where we can avoid polynomial dependence on $d$ in the rates. This is not a direct corollary of what is currently done in the paper, as the stochastic Frank-Wolfe algorithms of the references above are not clearly applicable to the online setting.

2. The issue about the smoothness constant $H$ is currently lacking a more thorough discussion.

**Strengths And Weaknesses:**

Strengths:

1. The unifying perspective on DP-SCO, capable to recover optimal rates without specialized arguments, or reductions between convex and strongly convex losses through regularization.
2. The perspective of online to batch conversions provides a rather straightforward extension to non-Euclidean norms, for which optimal methods are still scarce (Bassily et al. 2021).
3. Adaptivity to the noise level. All kinds of adaptivity in differential privacy are challenging, and not well understood, so this result in itself is significant.
4. The paper is fairly well written, and the proofs and analyses are not complicated and principled.

Weaknesses:

1. I think this unifying perspective becomes more interesting in geometries which are not $\ell_2$. More exploration of the consequences of the results of the paper to these settings would have been an interesting addition.
2. Aside from the noise adaptive rates and the parameter free algorithm, the rates derived in the paper are not new (namely, optimal rates have already been established). I don't think this is a major problem though, mostly because making these methods more broadly applicable is interesting and necessary to better understand DP-SCO.

---

> ### Author Response · Authors · 2022-08-01
> **Response to Reviewer ycjh**
>
> Thanks very much for your very careful and insightful review. We also appreciate your detailed line comments. We will address all of them to improve the final copy. Below we respond to a few of your more specific questions:
>
> **Q3:** It is true that $\beta_t\mathcal{L}(x_t) - g_t$ is a mean-zero random vector. However, the inner product $\langle \beta_t\mathcal{L}(x_t) - g_t, w_t - u \rangle$ is not mean-zero because $w_t$ is also a random vector that depends on $z_{1:t-1}$. For example, consider $\beta_t=1$ for all $t$ and the logistic loss $\ell(x, z) = \log(1+\exp(xz))$ for $z=\pm1$ with equal probability. Also choose OSD as the online learner which updates $w_{t+1} = w_t - \eta g_t$. Then following Algorithm 1 with $w_0=0, x_0=0$ as initialization, the expectation of the previous inner product in $t=2$ is $\eta/4$ instead of $0$.
>
> As a result, we need Holder’s inequality to bound this inner product with $D\\| \beta_t\mathcal{L}(x_t) - g_t\\|_*$ and further bound its expectation using Lemma 13 and 15. This is a subtle difference in our analysis compared to standard online-to-batch arguments.
>
> **Q4 and Q7:** Sorry for the confusion about the choice of $k$. The high level summary is: most of our results are easiest to prove with $k=1$, and it would indeed have been simpler to simply default to $k=1$ throughout the paper. However, for some results (e.g. Theorem 10), we require $k>1$ for some technical algebraic reasons, so we opted to prove the foundational theorems with general $k$ in case other future applications also benefit from $k>1$.
>
> The guiding principle behind working with $k$ is  the sensitivity of $\delta_t$ (line 5, Alg. 1). For $\beta_t = t^k$, the sensitivity of $\delta_t$ is of order $O(t^{k-1})$. For $k=1$, this is a constant sensitivity, which is particularly intuitive for analysis. For $k=0$ (i.e. the standard weighting in online-to-batch), the sensitivity is actually $O(1/t)$, which is much more complicated to analyze.  In order to apply the tree aggregation easily, we want the sensitivity of $\delta_t$ to be polynomial in $t$, rather than the inverse polynomial $1/t$, so we define $\beta_t = t^k$ and ask $k\geq1$.
>
> In all cases except for the parameter-free case, our results hold for any $k\geq1$, so we did not specify the value of $k$. However, in the parameter-free case, we further require $k>5/2$ for algebraic reasons, so we chose a specific value $k=3$. In the final copy, we will add more detailed discussion to eliminate confusion.
>
> **Q5:** Theorem 6 holds for any $k\geq 1$. The exponent $3/2$ implies that the term $Regret_T(x^*)/\beta_{1:T} = O(1/T^{3/2})$ is dominated by the other 2 terms in Theorem 4 of order $O(1/\sqrt{T} + 1/\rho T)$ so that the $O(1/\sqrt{T})$ term depends only on the unobserved true variances $\sigma_G,\sigma_H$.
>
> This implies the convergence rate in line 195. Compared to the convergence rate without the optimistic assumption in Remark 5, the optimistic rate is better because $\sigma_G < G$ if we assume $\ell$ is $G$-Lipschitz. In particular, if $\sigma_G \ll G$, the optimistic rate becomes significantly better.
>
> **Q6:** Sorry for the confusion, $R_i$ refers to the noise generated in line 12, Alg. 1.
>
> **Q8:** In Remark 5, the convergence rate is of order $O(\frac{D(G+D\sigma_H)}{\sqrt{T}} + \frac{\sqrt{d}D(G+DH)\log T}{\rho T})$, where $\sigma_H \leq H$. So, in the worst case, with $H=\sqrt{T}$, we might have $\sigma_H=\sqrt{T}$ and our bound becomes $O(1)$. In other words, we need to assume $H = o(\sqrt{T})$ to ensure a non-trivial bound from our algorithm.
>
> **Q12:** Theorem 19 and Appendix C is not used in the main text. We provide this section for interested readers who want to solve the privacy problem with pure $\epsilon$-DP instead of Renyi-DP (or equivalently $(\epsilon, \delta)$-DP). We can simply replace the $(V,\alpha)$-RDP distribution $\mathcal{D}$ in Algorithm 1 with a $V$-DP distribution (e.g., exponential distribution as proved in Theorem 19).
>
> **Q14:** Thanks for pointing out the recent reference [1] We believe that we do subsume some of their results (although not all). In particular, their Theorem 3.2 appears to be implied by our Theorem 4, and in particular remark 5. Notably we have greater generality along some axes by dint of our more general conversion.
>
> **Regarding L1 geometry:** You’re quite correct that it is not immediately obvious how to apply methods based on Frank Wolfe to our setup since these typically do not have as strong online learning regret bounds. We focused on the simpler L2 case and the immediate corollaries of our framework to adaptivity in this work to easily show off the value of our approach, but we think there is potential to extend the  results to L1 geometry in the future.
>
> **Reference**
> 1. Han, Y., Liang, Z., Liang, Z., Wang, Y., Yao, Y. and Zhang, J., 2022. On Private Online Convex Optimization: Optimal Algorithms in $\ell_p $-Geometry and High Dimensional Contextual Bandits. arXiv preprint arXiv:2206.08111.

---

> > ### Comment · Reviewer_ycjh · 2022-08-06
> > **Comments on Authors' Rebuttal**
> >
> > I would like to thank the authors for their detailed responses.
> >
> > Q3. I am still confused here. Right after Theorem 1 it says $\mathbb{E}[g_t|x_t]=\nabla {\cal L}(x_t)$. I would suppose it is also true that  $\mathbb{E}[g_t|w_1,\ldots,w_t]=\beta_t\nabla {\cal L}(x_t)$, hence
> > $$ \mathbb{E}( \mathbb{E}[\langle \beta_t \nabla {\cal L}(x_t)-g_t,w_t-u \rangle |w_1,\ldots,w_t]) )=
> >  \mathbb{E}(\langle \mathbb{E}[ \beta_t \nabla {\cal L}(x_t)-g_t|w_1,\ldots,w_t],w_t-u \rangle)  = 0 . $$
> > Please let me know if I am misunderstanding something here. BTW, I should say your analysis is correct. My only concern is whether it can be simplified.
> >
> > Q4 and Q7. It is interesting that somehow you benefit from a polynomially increasing sensitivity. I suppose in the end this effect is compensated by the tree aggregation and the averaging. This sounds that is something a more thorough discussion, so I support your idea of adding this to the paper.
> >
> > Q14. I am not exactly sure about the rules for papers being concurrent. But feel free to proceed as you consider more suitable.

---

> > > ### Author Response · Authors · 2022-08-07
> > > **Response to Reviewer ycjh**
> > >
> > > Thank you for your further comments. We respond to your comments regarding Q3 below:
> > >
> > > In regular online-to-batch where $g_t = \nabla\ell(w_t,z_t)$, it’s true that $\mathbb{E}[g_t | w_1,\ldots, w_t] = \nabla\mathcal{L}(w_t)$. The text right after Theorem 1 is discussing this more typical and familiar setting for $g_t$ rather than our somewhat different usage, but we agree the context switch is confusing and we will clarify in the final copy. In our case unfortunately it might not be that $\mathbb{E}[g_t | w_1,\ldots, w_t] = \beta_t\nabla\mathcal{L}(w_t)$.
> > >
> > > In our algorithm, we define $g_t$ in a tricky way: $g_t = \sum_{i=1}^t \beta_i\nabla\ell(x_i,z_i) - \beta_{i-1}\nabla\ell(x_{i-1}, z_t)$. Now, it is true that the *unconditional* expectation $\mathbb{E}[g_t- \beta_t \nabla \mathcal{L}(x_t)]$ is zero. However, our $g_t$ is no longer unbiased when conditioned on $w_1,\ldots, w_t$. The reason is that $w_1,\ldots, w_t$ are functions of $z_1,\ldots, z_{t-1}$, and $\mathbb{E}[g_t - \beta_t \nabla \mathcal{L}(x_t) | z_1,\dots,z_{t-1}]\ne 0$. Consider the following example:
> > >
> > > Let $\ell(x, z)=xz$ with $z=\pm1$ with equal probability. Consider the OSD algorithm with $w_{t+1} = w_t - \eta g_t$ (where $g_t$ is defined as in our algorithm) and $w_1 = 0$ as initialization. Let $\beta_0=0$ and $\beta_t=1$. In $t=1$, $w_1 = 0, x_1 = 0$, and $\delta_1 = \beta_1\nabla\ell(x_1,z_1) - \beta_0\nabla\ell(x_0,z_1) = z_1$, so $g_1 = \delta_1 = z_1$. In $t=2$, $w_2 = w_1 - \eta g_1 = -\eta z_1$, which is a function of $z_1$. Also, $\delta_2 = \beta_2\nabla\ell(x_2, z_2) - \beta_1 \nabla\ell(x_1, z_2) = z_2 - z_2 = 0$, and $g_2 = g_1 + \delta_2 = z_1$. Therefore, $\mathbb{E}[g_2 | w_1, w_2] = \mathbb{E}[z_1 | w_1=0, w_2 = -\eta z_1] =z_1\neq 0$. On the other hand, $\nabla\mathcal{L}(x) = 0$ for all $x$. That’s why we must use the more complicated bounding technique in Lemma 13 and 15.
> > >
> > > Using the same example, we can also show that $\beta_t = t$ is a better option. With $\beta_t=t$, we can check that $\delta_t = z_t$ and $g_t = z_1 + \dotsm + z_t$. Again, $g_t - \beta_t\nabla\mathcal{L}(x_t)$ is not mean-zero conditioned on $w_1,\ldots, w_t$ for the same reason (e.g., $w_2 = -\eta z_1$ and $g_2 = z_1+z_2$). However, compared to $g_t = z_1$ using $\beta_t = 1$, $g_t = z_1 + \dotsm + z_t$ is a better approximation of the actual weighted gradient $\beta_t \nabla\mathcal{L}(x_t) = 0$ (because even though now $|g_t|=O(\sqrt{t})$ is actually further from 0, we have upweighted everything by $\beta_t=t$.)

---

### Official Review · Reviewer_toms · 2022-07-08

**Rating:** 4
**Confidence:** 4
**Soundness:** 3 good
**Presentation:** 2 fair
**Contribution:** 2 fair

**Summary:**

This paper proposes a novel online-to-batch conversion algorithm with theoretical guarantees, which aims to achieve optimal convergence rate of expected risk under differential privacy constraints. Relationships between regret bound and expected risk are also provided under different conditions of loss function.

**Questions:**

What is the relationship between the proposed algorithm and existing differentially private online learning algorithms?

**Limitations:**

My major concern is about the reproducibility of the proposed algorithm. This paper proposes a novel online-to-batch algorithm with theoretical guarantees rather than only provides theoretical analyses for existing learning algorithms. Thus, empirical analyses are essential for demonstrating the effectiveness and efficiency of the novel algorithm with differential privacy constraints. Besides, some important related works about differentially private online learning are missing.

**Strengths And Weaknesses:**

Strengths:

1.	This work is well-motivated with theoretical analyses.

2.	Online-to-batch conversion technology is an important topic in online learning, and differentially private is also important in streaming data.

3.	Regret bound and expected risk are analyzed under different conditions of loss function.

Weaknesses:

1.	Some efforts have been made to the topic of differentially private online learning (DPOL) in [Jain et al., 2012]. The comparison with DPOL is missing.

2.	Lack of analysis of computational complexity.

3.	The paper lacks mostly discussion and insights for the theoretical results.

4.	This paper lacks empirical analyses, which limits the reproducibility of the proposed algorithms.

[Jain et al., 2012] Differentially private online learning, COLT 2012.

---

> ### Author Response · Authors · 2022-08-01
> **Response to Reviewer toms**
>
> Thank you very much for your constructive review. We respond to your specific comments below:
>
> 1. Regarding the comparison to private online learning [1]: Thank you for pointing out this related work. We want to emphasize that our paper solves a different problem than [1]. [1] converts non-private online learning algorithms into private online learning algorithms and its goal is regret minimization. On the other hand, our algorithm converts non-private online learning algorithms into private stochastic convex optimization (SCO) algorithms. In the SCO setup, there is one fixed loss function $\ell(x,z)$, and we aim to minimize the expected loss $\mathbb{E}_z[\ell(x,z)]$.
>
>     In other words, the setup in [1] is more difficult to solve, and their private online learning algorithm indeed can be converted into a private SCO algorithm via standard online-to-batch. However, their convergence rate is worse than ours: online-to-batch achieves convergence rate of order $O(\mathrm{Regret}_T(x^*)/T)$, and their regret is of order $\tilde O(\sqrt{T}/\epsilon)$ (Theorem 2, [1]), so the convergence rate is $\tilde O(1/\epsilon\sqrt{T})$. On the other hand, we achieve a better rate: $O(1/\sqrt{T} + 1/\epsilon T)$.
>
>     In the final copy, we will include this comparison.
>
> 2. Regarding the complexity analysis: In each round $t$, it takes $O(1)$ complexity in the main algorithm, and $O(\ln t)$ complexity in the $\text{Noise}$ subroutine. Overall, the complexity of the algorithm is $\tilde O(T)$.
>
> 3. Regarding theoretical results and insights: The main insight from our theoretical results is that we can convert any non-private online learning algorithms with $O(\sqrt{T})$ regret into private SCO algorithms with optimal convergence rate. In addition, our algorithm is a simple framework and can be easily modified for different scenarios. For example, in Section 4, we regularize the loss $\ell_t(w)$ sent to the online learner to achieve a faster convergence rate when the objective loss $\ell(w, z)$ is strongly convex.
>
>     Moreover, our algorithm inherits some properties from the online learning algorithms. In Section 3, we proved that if the online learner is optimistic, then our algorithm guarantees to converge faster. Similarly in Section 5, if the online learner is parameter-free, so is our algorithm. This is important because there is a vast literature on adaptive online learning, while adaptive methods are harder to come by in the private setting (as also noticed by Reviewer 3). We will add more detailed discussion about our theoretical insights in the final copy.
>
> 4. We agree that empirical experiments are important, and we plan to conduct an experimental study in the future. However, we feel our analytical results are original enough to be interesting in their own right: they provide a significantly expanded set of tools for building differentially private optimization algorithms.
>
> **Reference**
> 1. Jain, P., Kothari, P., & Thakurta, A. (2012). Differentially Private Online Learning. In Proceedings of the 25th Annual Conference on Learning Theory (pp. 24.1–24.34). PMLR.

---

### Official Review · Reviewer_HZ7s · 2022-07-21

**Rating:** 4
**Confidence:** 5
**Soundness:** 3 good
**Presentation:** 2 fair
**Contribution:** 2 fair

**Summary:**

Under the assumption of a smooth loss function, this paper investigates a new online-to-batch conversion method for private stochastic optimization.

**Questions:**

Is the cardinality of I_t an r.v. in Line 13 of Algorithm 1? If not, can we simplify the part involved I_t? If it is a r.v., can we use the randomness to amplify privacy?

**Ethics Review Area:**

["Privacy and Security (e.g., consent)"]

**Limitations:**

The writing could be improved to make it more reader-friendly, and experiments are encouraged..

**Strengths And Weaknesses:**

This paper offers a fresh look at stochastic optimization with DP guarantees. However, it appears to have been rushed and does not demonstrate well. Even though the smoothness assumption is a limitation, experiments for the smooth loss case and comparison with other existing methods are preferred to provide a sense of the utility advantage claimed in this paper.

---

> ### Author Response · Authors · 2022-08-01
> **Response to Reviewer HZ7s**
>
> Thank you very much for your constructive comments. We agree that empirical experiments will be helpful in comparing our algorithm to existing differentially private online convex optimization algorithms and we will conduct these experiments as our future work. Below we respond to your specific question:
>
> No, $I_t$ is a deterministic function of $t$ as defined in line 10-11. Intuitively, $I_t$ is the set of all required nodes in order to aggregate information for node $t$ in the tree aggregation algorithm [1].
>
> It is an interesting idea to attempt to inject noise into the choice of $I_t$. Note that even if there were some additional source of randomness here, we actually already obtain the optimal privacy guarantees, so it would be difficult to leverage it for any kind of amplification in the worst-case, but perhaps there is some additional adaptivity that could be obtained.
>
> 1. Dwork, C., Naor, M., Pitassi, T., and Rothblum, G. N. (2010). Differential privacy under continual observation. In Proceedings of the forty-second ACM symposium on Theory of computing, pages 715–724.

---

### Meta-Review · Area_Chair_PiUg · 2022-08-24

**Recommendation:** Accept
**Confidence:** Certain

**Metareview:**

The paper made original contributions to DP learning of convex and smooth differentially private learning problem by connecting to the vast parameter-free online learning literature. One of the reviewers read the paper  with great details and carefully checked the correctness of the results. The AC also took a close look and find the results very nice. The reviewers and the AC further discussed the work and clarified some of the concerns raised, e.g., regarding computation; but the missing experiments make it hard to vouch for the practicality of the approach.  Based on the theoretical contribution alone, we believe the paper is above the bar and would happily recommend accept.

The authors are encouraged to take into account the points below and consider adding benchmarking experiments.

---------------

Some additional feedback / comments out of the discussion:

1. For DP-SCO as the optimal rates are known to be achievable by a linear time algorithm (FKT'20). The proposed new algorithm thus does not improve over existing methods in either statistical or computational complexity.

2. If computation is not a concern, Noisy Gradient Descent with O(n^2) iterations (thus n^3 incremental gradient oracle calls) is known to provide an even stronger excess *empirical* risk bound that is optimal (without that additional log T in this paper). It also is very hard to beat in practice.

3. The main contribution of the proposed algorithm is then about new algorithmic techniques borrowed from adaptive online learning. This approach gives rise to the unified treatment of general convex and strongly convex problems and they discussed how it helps to tap into other more adaptive / more parameter-free online learners.

4.  Tree-aggregation for releasing gradient sequences by leveraging smoothness (and stability implied by the anytime online-to-batch reduction) is cute. I think technically this is the most interesting idea.

5. I think the paper contains enough good results to be considered as a purely theoretical work for acceptance. That said, I do think the algorithm has the potential to be practical. It is a pity that the authors did not try. Though with the several layers of reductions and the binary-tree approach, it won't surprise me if the proposed method is not competitive against methods such as NoisyGD or NoisySGD.

6. Another positive aspect about this paper is that it is polished and the writing is pretty good. I particularly liked how the authors accurately describe their contribution in the title / abstract by clearly highlighting the key assumption on the smoothness.




**Award:**

No

---

### Decision · Program_Chairs · 2022-09-14

Accept